# Improving Self-supervised Molecular Representation Learning using Persistent Homology

**Yuankai Luo**
Beihang University
`luoyk@buaa.edu.cn`

**Lei Shi**
Beihang University
`leishi@buaa.edu.cn`

**Veronika Thost**
MIT-IBM Watson AI Lab,
IBM Research
`veronika.thost@ibm.com`

## Abstract

Self-supervised learning (SSL) has great potential for molecular representation learning given the complexity of molecular graphs, the large amounts of unlabelled data available, the considerable cost of obtaining labels experimentally, and the hence often only small training datasets. The importance of the topic is reflected in the variety of paradigms and architectures that have been investigated recently, most focus on designing views for contrastive learning. In this paper, we study SSL based on persistent homology (PH), a mathematical tool for modeling topological features of data that persist across multiple scales. It has several unique features which particularly suit SSL, naturally offering: different views of the data, stability in terms of distance preservation, and the opportunity to flexibly incorporate domain knowledge. We (1) investigate an autoencoder, which shows the general representational power of PH, and (2) propose a contrastive loss that complements existing approaches. We rigorously evaluate our approach for molecular property prediction and demonstrate its particular features in improving the embedding space: after SSL, the representations are better and offer considerably more predictive power than the baselines over different probing tasks; our loss increases baseline performance, sometimes largely; and we often obtain substantial improvements over very small datasets, a common scenario in practice.

## 1 Introduction

Self-supervised learning (SSL) has great potential for molecular representation learning given the complexity of molecules, the large amounts of unlabelled data available, the considerable cost of obtaining labels experimentally, and the resulting often small datasets. The importance of the topic is reflected in the variety of paradigms and architectures that are investigated [Xia et al., 2023a].

Most existing approaches use *contrastive learning* (CL) as proposed in [You et al., 2020]. CL aims at learning an embedding space by comparing training samples and encouraging representations from positive pairs of samples to be close in the embedding space while representations from negative pairs are pushed away from each other. However, usually, each given molecule is considered as its own class, that is, a positive sample pair consists of two different views of the same molecule; and all other samples in a batch are used as the negative pairs during training. We observe that this represents a very coarse-grained comparison basically ignoring all commonalities and differences between the given molecules. Therefore also current efforts in molecular CL are put in constructing views that capture the possible relations between molecules: **GraphCL** [You et al., 2020] proposes four

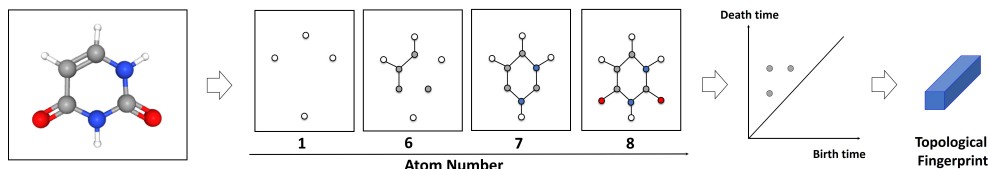

Figure 1: Topological fingerprints are constructed using a filter (e.g., atom number) and recording resulting topological structures in a PD which is then vectorized.

simple augmentations (e.g., drop nodes), but fix two per dataset; **JOAO** [You et al., 2021] extends the former by automating the augmentation selection; **GraphLoG** [Xu et al., 2021] optimizes in a local neighborhood of the embeddings, based on similarity, and globally, using prototypes based on semantics; and **SimGRACE** [Xia et al., 2022] creates views using a second encoder, a perturbed version of the molecule's ones; also most recent works follow this paradigm [Wu et al., 2023].

**Quality of Embedding Spaces is Underinvestigated.** Although differences in performance are often observed, they seem marginal and are not yet fully comprehended. With a few exceptions, the models are usually evaluated over the MoleculeNet benchmark [Wu et al., 2018a] only, a set of admittedly rather diverse downstream tasks. While those certainly provide insights into model performance, recent evaluation papers have pointed out that the picture may be very different when the models are evaluated in more detail [Sun et al., 2022, Wang et al., 2022a, Akhondzadeh et al., 2023, Deng et al., 2022]. In particular, [Akhondzadeh et al., 2023] propose to use linear probing to analyse the actual power of the representations instead of a few, specific downstream tasks. This type of evaluation is also common in other areas of DL [Chen et al., 2020]. [Deng et al., 2022] consider other, smaller datasets, and compare to established machine learning (ML) approaches.

**We study molecular SSL based on persistent homology (PH).** PH is a mathematical tool for modeling topological features of data that persist across multiple scales. In a nutshell, the molecule graph can be constructed sequentially using a custom filter, such that atom nodes and the corresponding bond edges only appear once they meet a given criterion (e.g., based on atomic mass); see Figure 1. PH then captures this sequence in a persistence diagram (PD), and the area has developed various methods to translate these diagrams into *topological fingerprints*, which can be used for ML [Ali et al., 2022].

Topological fingerprints have been used for *supervised* ML over molecules by chemists [Krishnapriyan et al., 2021, Demir et al., 2022] and recent work in AI has shown promising results compared to the commonly used ECFP embeddings [Rogers and Hahn, 2010]. Moreover, many of these fingerprints have been shown to be *stable*, in the sense that distances between the PDs are reflected in the distances between the corresponding fingerprints. *We point out that the unique features of topological fingerprints particularly suit molecular SSL*, naturally offering: different views of the data (i.e., by switching the filter function), stability in terms of distances, and the opportunity to flexibly incorporate domain knowledge.

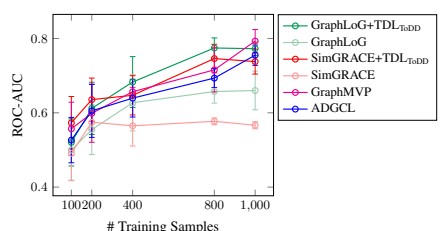

Figure 2: TDL may yield great improvements in the low-data scenario; ClinTox.

**Our Idea.** We consider the topological fingerprints of the molecules in the pre-training data as views and exploit their stability to model the distances between the *given* molecules (i.e., instead of between a molecule and/or its views). In particular, we use them for fine-grained supervision in SSL.

**Contributions.**

- We propose a **topological distance contrastive loss (TDL)** which, as outlined above, shapes the embedding space by providing supervision for the relationships between molecules, especially the ones in the data. TDL is complementary and *can be flexibly and efficiently applied to complement any CL approach*. We extensively evaluate it combining TDL with four established CL models.
- We also consider *alternative paradigms for comparison*: we study a straightforward **topological fingerprints autoencoder (TAE)**.
- Our evaluation particularly focuses on the *potential in improving the embedding space* and we demonstrate the gain in representation power in detail empirically (see Section 4.1): TDL en-

ables the models to learn, in a sense, calibrated distances; considerably improves linear probing performance; and the fine-grained supervision may mitigate deficiencies of individual models.
- Over downstream data (see Section 4.2) the performance increases depend more on the model and data. Notably, TDL is able to considerably improve GraphCL and make it competitive with SOTA. Moreover, TDL *strongly improves various models in the low-data scenario* (see Figure 2).

Our implementation is available at `https://github.com/LUOyk1999/Molecular-homology`.

## 2    Background and Related Works

**Graph Homology.** Molecules are graphs $G = (V, E)$ with nodes $V$, the atoms, and bond edges $E$. In algebraic topology, graph homology considers such a graph $G$ as a topological space. We focus on *simplices*: every node is a 0-simplex, and every edge is a 1-simplex. In the context of graphs, we obtain a 1-dimensional *simplicial complex* $X = V \cup E$ in an easy way, by considering the simplices induced by $G$; the dimension is determined by the maximal dimension of the contained simplices.

**Persistent Homology (PH).** We introduce the most important concepts and, for more details, refer to [Dey and Wang, 2022, Edelsbrunner and Harer, 2022]. Persistent homology is a mathematical tool for modeling topological features of data that persist across multiple scales, comparable to different resolutions. These features are captured in persistent diagrams which, in turn, can be vectorized in fingerprints. We outline the process simplified below and in Figure 1; see Appendix A for details.

First, the goal is to construct a nested sequence of subgraphs $G_1 \subseteq ... \subseteq G_N = G$ $(1 \leq i \leq N)$. As described above, these graphs can be considered as simplicial complexes, hence we have simplices which we can record in a persistence diagram. To this end, we consider one of the most common types of filtration methods: sublevel/superlevel filtrations. A *sublevel filtration* is a function $f : X \to \mathbb{R}$ over all simplices. A simple such function $f$ can be a node-valued function (e.g., map atom nodes to their atomic number) that is expanded to the edges as $f(u, v) = max(f(u), f(v))$. Denote by $X_a$ the sublevel set of $X$, consisting of simplices whose filtration function values $\leq f(a)$, $X_a = \{x \in X | f(x) \leq f(a)\}$. As the threshold value $a$ increases from $\min_{v \in V} f(v)$ to $\max_{v \in V} f(v)$, let $G_a$ be the subgraph of $G$ induced by $X_a$; i.e., $G_a = (V_a, E_a)$ where $V_a = \{v \in X_a\}$ and $E_a = \{e_{rs} \in X_a\}$. This process yields a nested sequence of subgraphs $G_1 \subseteq G_2 \subseteq ... \subseteq G_N = G$. As $X_a$ grows to $X$, new topological structures gradually appear (born) and disappear (die).

Second, a *persistence diagram* (PD) is obtained as follows. For each topological structure $\sigma$, PH records its first appearance and its first disappearance in the filtration sequence. And this is represented by a unique pair $(b_\sigma, d_\sigma)$, where $1 \leq b_\sigma \leq d_\sigma \leq N$. We call $b_\sigma$ the birth time of $\sigma$, $d_\sigma$ the death time of $\sigma$ and $d_\sigma - b_\sigma$ the persistence of $\sigma$. PH records all these birth and death times of the topological structures in persistence diagram $PD(G) = \{(b_\sigma, d_\sigma) | \sigma \in H_k(G_i)\}$, where $H_k(G_i)$ denotes the $k$-th homology group of $G_i$, and in practice, $k$ typically takes values 0 or 1. This step is rather standard and corresponding software is available [Otter et al., 2017].

**PH in ML.** The vectorization of PDs for making them applicable in ML has been studied extensively [Ali et al., 2022], and proposals range from simple statistical descriptions to more complex *persistence images* (PIs) [Adams et al., 2017]. In a nutshell, for PIs, the PD is considered as a 2D surface, tranformed using a Gaussian basis function, and finally discretized into a vector. Furthermore, the Euclidean distance between PIs is *stable* with respect to the 1-Wasserstein distance between PDs [Adams et al., 2017], which essentially means that the former is bounded by a constant multiple of the latter. We focus on PIs based on the promising results reported in supervised settings [Demir et al., 2022, Krishnapriyan et al., 2021]. Our study was inspired by the ToDD framework, applying pre-trained vision transformers to custom 2D topological fingerprints [Demir et al., 2022]. Interestingly, they use the distances between the transformer's molecule representations to construct a suitable dataset for subsequent *supervised* training using triplet loss. Our focus is on SSL and we apply the ToDD fingerprints as more complex, expert fingerprints in comparison to the ones based on atomic mass only. Recently, various other approaches of integrating PH into ML are explored, but these are only coarsely related to our work (e.g., [Horn et al., 2022, Yan et al., 2022]).

**Molecular SSL.** Since the foundational work of [Hu* et al., 2020], who have proposed several effective methods, such as the node context prediction task ContextPred, the area is advancing at great pace [Xia et al., 2023a, Xie et al., 2022]. Our proposal falls into the category of contrastive learning as introduced in Section 1. Our study focuses on the potential of *PH to complement existing*

*models* such as [You et al., 2020] and follow-up approaches. There are other related CL approaches to which we do not aim to compare to directly; e.g., works including 3D geometry [Liu et al., 2022a, Stärk et al., 2022] or considerably scaling up the pre-training data [Ross et al., 2022, Zhou et al., 2023]. In contrast, our focus is on improvement through exploiting unused facets of the data.

**SSL based on Distances.** We found only few works that explicitly incorporate distances into SSL. With regard to graphs, [Kim et al., 2022] exploit the graph edit distance between a graph and views to similarly represent the actual distance in the embedding space inside the loss function (i.e., vs. invariance to the view transformation). This distance computation is feasible since it can be easily obtained based on the transformation. Our work focuses on exploring in how far distances in terms of PH can be applied towards the same goal; moreover, this makes it feasible to model the distances between given samples. [Wang et al., 2022b] consider a loss very similar to our proposal based on the more coarse-grained ECFPs, instead of PIs, but they focus on chemistry aspects instead of SSL more generally. Also [Zha et al., 2022] model distances between given samples in the context of CL in a similar way, but in the context of a supervised scenario, where the distances are the differences between given regression labels; note that they also list other coarser related works. [Taghanaki et al., 2021] apply custom distances inside a triplet loss but similarly exploit label information, in order to select the positive and negative samples. Beyond that, distances have been applied to obtain hard negatives [Zhang and Re, Demir et al., 2022], and are exploited in various other ways rather different from our method. For instance, several recent approaches aim to structure the embedding space by exploiting correlations between samples which are, in turn, obtained using nearest neighbors [Caron et al., 2020, Dwibedi et al., 2021, Ge et al., 2023]. Also methods considering equivariance between view transformations implicitly model distances [Chuang et al., 2022, Devillers and Lefort, 2023].

**Others.** While our focus on CL and distances hints at a close relationship to deep metric learning [KAYA and BİLGE, 2019], these works usually do not have explicit distances for supervision but, for instance, exploit labels. Observe that this indicates the unique nature of the distances PH provides us with. Lastly, SSL more generally [Rethmeier and Augenstein, 2023] has naturally been inspiration for molecular SSL and is certainly one reason why the field was able to advance so fast. We use its insights by putting focus on linear probing and investigating dimensional collapse [Hua et al., 2021].

# 3 Methodology

The main goal of SSL is to learn an embedding space that faithfully reflects the complexity of molecular graphs, captures the topological nature of the molecular representation space overall, and whose representations can be effectively adapted given labels during fine-tuning. We propose methods based on persistent homology and show that they naturally suit SSL. First, persistence images (or comparable topological fingerprints) offer great versatility in that they are able to flexibly represent knowledge about the graphs and allow for incorporating domain knowledge. Second, they capture this knowledge based on persistence diagrams, which is very different from - and hence likely complementary to - the common graph representation methods in deep learning. Third, and most importantly, their stability represents a unique feature, which makes them ideal views for SSL.

We study two SSL approaches based on PH and evaluate them in detail in terms of both their impact on representation power (see Section 4.1) and on downstream performance (see Section 4.2):

- In order to study the impact of PH on SSL in general, we consider a simple *autoencoder* architecture.
- Since we consider topological fingerprints to represent information that is complementary to that used in existing approaches (we also show that they are not ideal alone) and because their topological nature can be used to improve the latter in unique ways, we developed a loss function based on *contrastive learning* that complements existing approaches.

In this paper, our focus is on obtaining initial insights about the potential PH offers for molecular SSL, hence we chose one basic solution and one providing unique impact. There are certainly other promising ways to be investigated in the future.

## 3.1 Topological Fingerprints AutoEncoder (TAE)

Autoencoders are designed to reconstruct certain inputs given context information for the input graph $G$. We consider topological fingerprints $I_G$ as the reconstruction targets, specifically, PIs.

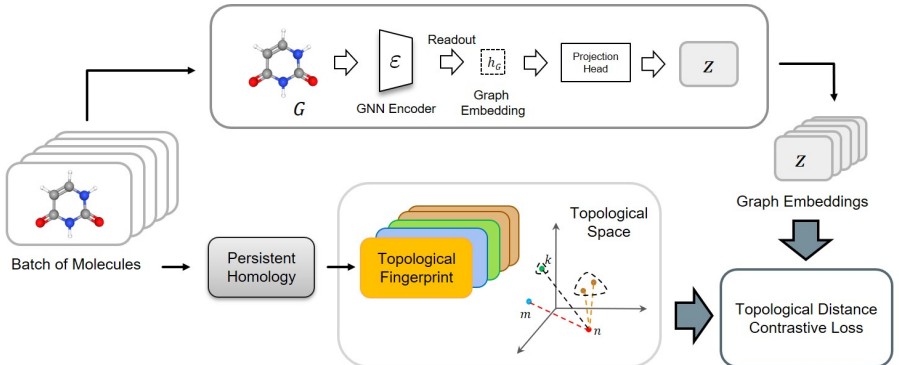

Figure 4: Overview of Topological Distance Contrastive Learning.

The model itself employs a typical graph encoder $\varepsilon(\cdot)$ for computing embeddings $h_v$ for the individual nodes $v \in V$; for simplicity, we write $h_V = \varepsilon(G)$, where $h_V = \{h_v | v \in V\}$ represents all node representations. Next, we pass it through a projection head $g(\cdot)$ and readout function $R(\cdot)$ (e.g., mean pooling) to obtain the graph-level representation $h_G = R(g(h_V))$. Hence the context is the particular knowledge the graph encoder was de-

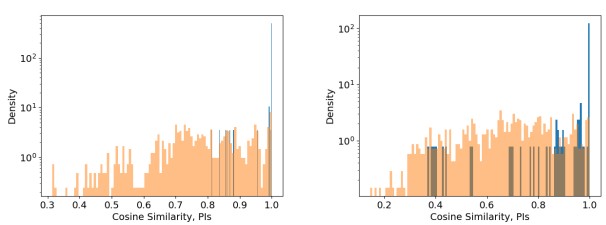

Figure 3: Comparison of molecule similarity based on PIs to similarity between corresp. ECFPs (blue: 20% most similar).

signed to exploit; in our evaluation, we consider the commonly used GIN [Xu et al.] in order to ease comparison with related works. As our loss formula for this *topological fingerprints autoencoder* (TAE) we use the regular mean squared error: $\mathcal{L}_{\text{TAE}} = \sum_G \text{MSE}(h_G, I_G)$.

**Observations.** Given stable fingerprints such as PIs, one main usual criticism for autoencoders, the fact that they fail to capture inter-molecule relationships, does not apply if the model is able to reliably learn the PIs, which we show TAE does; in particular, we observe a strong correlation between PIs and their reconstructions (see Table 7). The successful application of topological fingerprints in supervised learning and the simplicity of the model justify the study of TAE for analysis, however, we note that the PIs employed may not necessarily capture all information which is critical for a particular downstream task. For example, Figure 3 shows that PIs may offer a generally fitting embedding space, in that randomly chosen molecule pairs with similar standard fingerprints based on substructures (blue, Tanimoto similarity of ECFPs) have rather similar PIs (x-axis, cosine similarity). However, there is a clear difference for the two depicted datasets, sometimes PIs as shown here, based on the ToDD filtration Demir et al. [2022], fail to fully capture structural similarity; and this is directly reflected in performance (see Table 4). While optimization on a case-by-case basis w.r.t. the choice of fingerprints is possible, this is not in the spirit of foundational SSL, requires expert knowledge or extensive tuning, and may still not be sufficient. For that reason, we suggest to apply them together with existing approaches and show that they offer unique benefits.

## 3.2 Topological Distance Constrastive Loss (TDL)

Contrastive learning aims at learning an embedding space by comparing training samples and encouraging representations from positive pairs of examples to be close in the embedding space while representations from negative pairs are pushed away from each other. Current approaches usually consider each sample as its own class, that is, a positive pair consists of two different views of it; and all other samples in a batch are used as the negative pairs during training. We observe that this represents a very coarse-grained comparison basically ignoring all commonalities and differences between the given molecules; this is also why current efforts in graph CL focus on constructing views that capture the possible relations between molecules.

**Our Idea, Figure 4.** We exploit the stability of topological fingerprints such as PIs to model the distances between the *given* molecules (i.e., instead of just views of the same molecule) and to use them for fine-grained supervision in SSL (recall that stability means the distances between PIs reflect those between the topological persistence diagrams). This is very different from and complementary to related works in that it *structures the embedding space in a different way*. The fingerprints are usually constructed in a way so that they capture information about the molecular graph structure; even if they do not capture the entire complexity of the molecules, they represent some, probably important aspects. Moreover, based on the topological nature, they *may capture aspects not represented by the commonly used graph embeddings*. In particular, they offer a *way to flexibly integrate expert knowledge* into molecular SSL; observe that the usually employed GNNs have dimensions based on the available molecule information, hence including additional information, even if available only for some of the data, requires architecture changes.

We consider a batch of $N$ graphs, $\{G_i\}_{i \in [1,N]}$. Similar as above, we first extract graph-level representation vectors $h_{G_i}$ using a graph encoder $\varepsilon(\cdot)$, followed by a readout function $R(\cdot)$. Here, we apply the projection head $g(\cdot)$ later, to map the graph representations to another latent space and obtain the final graph embedding $z_i$. Specifically, we apply a two-layer MLP and hence a non-linear transformation, known to enhance performance [Chen et al., 2020]: $z_i = g\left(R\left(\varepsilon\left(G_i\right)\right)\right)$

Let $G_n$ be the sample under consideration. Instead of constructing an artificial view, we consider all possible positive pairs of samples $(G_n, G_m)$ together with a set of stable, topological fingerprints $\{I_i\}_{i \in [1,N]}$ for all graphs. Note that these can be computed rather efficiently, and they have to be computed only once for the given pre-training data. In a nutshell, our loss adapts the regular NT-Xent [Sohn, 2016, Oord et al., 2018, Wu et al., 2018b] by considering only those negative pairs $(G_n, G_k)$ of samples where the Euclidean distance $dis(I_n, I_k)$ between the corresponding fingerprints is greater than the one between $I_n$ and $I_m$. We compute the similarity score as $sim(z_n, z_m) = z_n^\top z_m / \|z_n\| \|z_m\|$, and consider a temperature parameter $\tau$ and indicator function $\mathbb{I}_{[\cdot]} \in \{0,1\}$ as usual. Our *topological distance contrastive loss* (TDL) is defined as follows, for the n-th sample:

$$\mathcal{L}_{\text{TDL}_n} = \frac{1}{N-1} \sum_{\substack{m \in [1,N], \\ m \neq n}} -\log \frac{e^{sim(z_n, z_m)/\tau}}{\sum_{\substack{k \in [1,N], \\ k \neq n}} \mathbb{I}_{[dis(I_n, I_k) \geq dis(I_n, I_m)]} \cdot e^{sim(z_n, z_k)/\tau}}$$

TDL *can be flexibly and efficiently applied to complement any graph CL framework*, e.g., in the form $\mathcal{L}_n = \mathcal{L}_{others} + \lambda \mathcal{L}_{\text{TDL}_n}$, where $\lambda$ determines its impact. In our evaluation, we used $\lambda = 1$.

**Further Intuition.** Essentially, TDL provides a form of regularization. It encourages the model to push molecule representations less far away from the sample under consideration if they are similar to it in terms of the topological fingerprints. Given the stability of those, we hence obtain an embedding space with better calibrated distances (i.e., distances in terms of PH, between persistence diagrams of the molecule graphs). This can be partly observed theoretically, in the directions of the gradients; see Appendix I for an initial analysis. We show this empirically by calculating the correlation between distances between the molecule representations after pre-training and the PIs (see Table 1), and by visualizing the distances in Figure 5. Our evaluation further shows that the fine-grained supervision may solve deficiencies of CL models in that it forces them to capture crucial features of the input. Further, the improved embedding space particularly suits low-data downstream scenarios.

**On Views.** While we consider the modeling of the sample relationships to be most unique and offer great potential to complement other models, we note that TDL can be similarly applied over views.

# 4 Evaluation

- Do TAE and TDL lead to, in a sense, **calibrated distances in the representation space**?
- Do we obtain **improved representations** more generally, based on established SSL metrics?
- What impact do we see on **downstream performance**, and does it justify our proposal?

**Our Models.** We apply both TAE and TDL with different filtration functions, marked by a subscript. First, we apply a most simple filtration `atom` based on atomic mass. This allows showing that, even by considering less information than the baseline GNNs, which additionally apply atom chirality and connectivity, topological modeling may exploit additional facets of the data. Since TAE is a standalone model, atomic mass is not enough to let it fully capture the molecular nature. For that

Table 1: Pearson correlation coefficients (%) between distances in embedding space and distances between corresponding PIs, for samples from various MoleculeNet datasets. Highlighted are clear decreases and clear increases (i.e., considering standard deviation).

|  | Tox21 | ToxCast | Sider | ClinTox | MUV | HIV | BBBP | Bace |
|---|---|---|---|---|---|---|---|---|
| ContextPred | 31.2 (0.4) | 2.5 (0.0) | 61.6 (0.2) | 15.6 (0.5) | 37.2 (0.1) | 20.6 (0.1) | 3.7 (0.3) | 12.7 (0.2) |
| + TAE$_{ahd}$ | 35.6 (0.3) | 12.2 (0.6) | 60.6 (0.3) | 6.2 (1.6) | 55.9 (0.1) | 30.9 (0.1) | 2.7 (0.7) | 25.9 (0.2) |
| GraphCL | 15.6 (0.4) | 7.4 (0.8) | 52.6 (0.2) | 16.7 (0.8) | 35.4 (0.2) | 18.3 (0.2) | 6.2 (1.8) | 10.5 (0.2) |
| + TDL$_{atom}$ | 55.4 (0.2) | 14.5 (0.4) | 66.6 (0.3) | 21.9 (0.7) | 65.2 (0.3) | 41.5 (0.1) | 7.0 (1.4) | 34.1 (0.3) |
| JOAO | 25.1 (0.8) | 3.2 (1.7) | 62.4 (0.3) | 20.2 (1.3) | 48.0 (0.2) | 29.5 (0.3) | 2.7 (1.4) | 24.8 (0.3) |
| + TDL$_{atom}$ | 49.3 (0.3) | 14.6 (2.1) | 65.2 (0.1) | 26.2 (1.0) | 60.2 (0.2) | 40.6 (0.4) | 6.7 (0.8) | 36.0 (0.4) |
| SimGRACE | 10.3 (2.9) | 9.8 (0.6) | 59.9 (1.1) | 4.8 (1.6) | 26.5 (0.3) | 21.6 (0.2) | 9.5 (2.1) | 4.4 (0.1) |
| + TDL$_{atom}$ | 48.8 (0.9) | 12.8 (1.7) | 61.5 (0.2) | 20.2 (0.5) | 71.9 (0.1) | 49.1 (0.2) | 8.5 (2.0) | 38.9 (0.1) |
| GraphLoG | 16.8 (0.2) | 2.9 (0.5) | 34.2 (0.2) | 3.6 (0.8) | 20.4 (0.1) | 9.4 (0.2) | 3.3 (0.9) | 12.6 (0.2) |
| + TDL$_{atom}$ | 44.2 (0.4) | 11.8 (1.3) | 50.5 (0.4) | 22.4 (1.2) | 63.9 (0.1) | 46.2 (0.2) | 1.9 (0.4) | 40.3 (0.1) |

reason, we consider three filtrations (atomic mass, heat kernel signature, node degree) and concatenate the corresponding PIs, denoted by `ahd`. Finally, to show the real potential of PH, we include the `ToDD` filtration Demir et al. [2022], which combines atomic mass with additional domain knowledge, partial charge and bond type, inside a more complex multi-dimensional filtration. See Appendix A.

**Baselines & Datasets.** For a comprehensive evaluation, we apply TDL on a variety of existing CL approaches: GraphCL, JOAO, GraphLoG, and SimGRACE (see Section 1). Note that this goes far beyond other CL extensions which are often evaluated with a single approach only [You et al., 2021, Xia et al., 2023b]. We also study TAE on top of the established ContextPred [Hu* et al., 2020], to get an idea of its complementary nature. Model configurations and experimental settings are described in Appendix A. For pre-training, we considered the most common dataset following [Hu* et al., 2020], 2 million unlabeled molecules sampled from the ZINC15 database [Sterling and Irwin, 2015]. For downstream evaluation, we focus on the MoleculeNet benchmark [Wu et al., 2018a] here, the appendix contains experiments on several other datasets.

### 4.1 Analysis of Representations after Pre-training

**Calibrated Distances in Embedding Space, Tables 1, 7, 9, Figure 5.**

TAE successfully learns the PI space for the molecules in MoleculeNet in the sense that there is a strong correlation between the PIs and their reconstructions (Table 7, appendix). For TDL, we observe for all baselines considerable increases in terms of correlation between distances in PI space and in representation space; we use the embeddings from linear probing. There are two notable exceptions and generally smaller increases on one dataset, BBBP. We also compared the ROGI score [Aldeghi et al., 2022] which, in a nutshell, measures in how far (dis)similarity of molecules (i.e., we use Euclidean distance of embeddings) is reflected in label similarity (Table 9, appendix). This gives us some idea in terms of downstream labels and the mixed results reflect the variety of the data. Finally, the alignment figure on the right clearly visualizes that the PI embedding space is much more fine-grained in terms of distances than the GNN space, and that the distribution of GraphCL seems to get correctly adapted in that distances between "positive" pairs generally shrink below the ones of "negative" pairs.

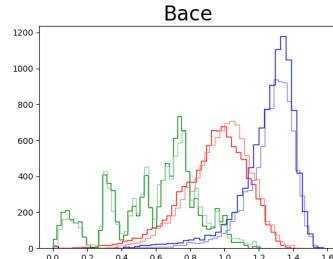

Figure 5: Normalized Euclidean distances for pairs of embeddings after pre-training of same/different Bace class, dark/light: PI$_{ToDD}$, green; GraphCL, blue; GraphCL+TDL$_{ToDD}$, red.

**Mitigating Dimensional Collapse, Figure 6.** One of the most interesting of our findings is the fact that GraphCL and GraphLoG suffer from dimensional collapse (i.e., the embeddings span a lower-dimensional subspace instead of the entire available embedding space, [Wang et al., 2022a] also observe this for GraphCL) and that TDL successfully mitigates this. This can be observed in

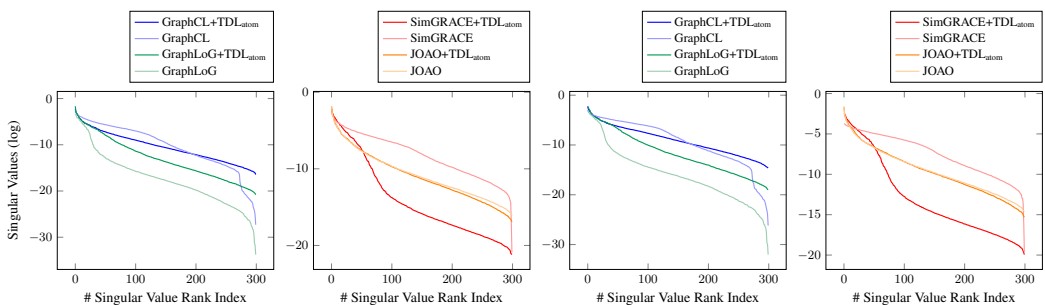

Figure 6: Singular values of covariance matrices of the representations; Bace (left), Clintox (right).

Table 2: Linear/MLP probing: molecular property prediction; binary classification, ROC-AUC (%).

| | Tox21 | ToxCast | Sider | ClinTox | MUV | HIV | BBBP | Bace | Average |
|---|---|---|---|---|---|---|---|---|---|
| $PI_{atom}$ | 56.9 (0.3) | 51.5 (0.4) | 56.1 (0.5) | 45.5 (1.1) | 58.6 (0.7) | 68.4 (0.6) | 47.1 (0.4) | 48.6 (0.7) | 54.08 |
| $PI_{ToDD}$ | 65.8 (0.3) | 50.3 (0.3) | 58.1 (0.5) | 56.9 (1.3) | 55.7 (0.9) | 70.8 (0.4) | 57.1 (0.7) | 67.8 (0.8) | 60.31 |
| ECFP | 68.8 (0.3) | 57.0 (0.2) | 62.3 (0.3) | 69.2 (0.8) | 66.1 (0.4) | 69.4 (0.2) | 63.2 (0.3) | 73.6 (0.8) | 66.20 |
| ECFP ‖ $PI_{ToDD}$ | 69.6 (0.4) | 56.3 (0.3) | 60.9 (0.6) | 76.7 (1.1) | 64.0 (0.6) | 71.6 (0.5) | 63.0 (0.4) | 76.8 (1.0) | **67.36** |
| $PI_{atom}$, MLP | 57.2 (0.5) | 52.5 (0.4) | 56.6 (0.7) | 49.8 (1.4) | 60.5 (1.6) | 69.9 (0.4) | 48.8 (1.0) | 53.3 (1.1) | 56.08 |
| $PI_{ToDD}$, MLP | 66.7 (0.3) | 52.5 (0.4) | 58.6 (0.6) | 61.8 (1.6) | 60.1 (0.4) | 71.6 (0.7) | 57.3 (0.9) | 68.2 (1.3) | 62.03 |
| ECFP, MLP | 70.1 (0.4) | 59.8 (0.4) | 59.6 (0.6) | 67.8 (0.9) | 61.7 (0.8) | 69.1 (1.0) | 58.6 (1.3) | 72.1 (1.7) | 64.85 |
| ECFP ‖ $PI_{ToDD}$, MLP | 71.1 (0.6) | 57.8 (0.4) | 59.2 (0.7) | 80.7 (2.1) | 64.9 (1.1) | 72.8 (1.7) | 63.1 (0.8) | 76.7 (0.9) | **68.28** |
| $TAE_{ahd}$ | 67.7 (0.2) | 61.2 (0.2) | 55.8 (0.3) | 58.1 (0.7) | 70.2 (0.8) | 72.5 (0.5) | 61.1 (0.2) | 74.3 (0.2) | 65.11 |
| $TAE_{ToDD}$ | 70.4 (0.2) | 60.8 (0.1) | 61.1 (0.1) | 68.4 (0.7) | 72.3 (0.3) | 73.9 (0.2) | 61.6 (0.4) | 67.6 (0.6) | 67.01 |
| ContextPred | 68.4 (0.3) | 59.1 (0.2) | 59.4 (0.3) | 43.2 (1.7) | 71.0 (0.7) | 68.9 (0.4) | 59.1 (0.2) | 64.4 (0.6) | 61.69 |
| + $TAE_{ahd}$ | 69.7 (0.1) | 59.2 (0.2) | 59.5 (0.3) | 56.1 (1.1) | 76.5 (0.9) | 68.9 (0.2) | 61.1 (0.4) | 65.6 (0.5) | **64.58** |
| + $TAE_{ToDD}$ | 69.0 (0.1) | 59.8 (0.4) | 60.0 (0.4) | 53.3 (1.3) | 70.8 (0.3) | 70.0 (0.7) | 60.9 (0.5) | 62.7 (0.5) | 63.31 |
| GraphCL | 64.4 (0.5) | 59.4 (0.2) | 54.6 (0.3) | 59.8 (1.2) | 70.2 (1.0) | 63.7 (2.3) | 62.4 (0.7) | 71.1 (0.7) | 63.20 |
| + $TDL_{atom}$ | 72.0 (0.4) | 61.1 (0.2) | 59.7 (0.6) | 65.3 (1.3) | 76.1 (0.9) | 68.2 (1.1) | 65.4 (0.9) | 76.4 (1.1) | **68.02** |
| + $TDL_{ToDD}$ | 72.7 (0.5) | 60.8 (0.4) | 58.9 (0.8) | 64.1 (1.7) | 72.7 (1.4) | 69.7 (1.2) | 64.5 (0.8) | 76.1 (1.3) | 67.44 |
| JOAO | 70.6 (0.4) | 60.5 (0.3) | 57.4 (0.6) | 54.1 (2.6) | 69.8 (1.9) | 68.1 (0.9) | 63.7 (0.3) | 71.2 (1.0) | 64.42 |
| + $TDL_{atom}$ | 70.5 (0.3) | 60.4 (0.2) | 57.8 (1.5) | 54.6 (1.3) | 74.2 (1.6) | 68.2 (0.6) | 65.2 (0.3) | 72.7 (3.1) | **65.41** |
| + $TDL_{ToDD}$ | 71.7 (0.4) | 61.3 (0.3) | 58.9 (0.7) | 52.4 (1.7) | 69.6 (1.7) | 69.9 (0.6) | 64.1 (0.5) | 72.6 (0.9) | **65.06** |
| SimGRACE | 64.6 (0.4) | 59.1 (0.2) | 54.9 (0.6) | 63.4 (2.6) | 67.4 (1.2) | 66.3 (1.5) | 65.4 (1.2) | 67.8 (1.3) | 63.61 |
| + $TDL_{atom}$ | 68.6 (0.3) | 61.1 (0.2) | 59.5 (0.4) | 62.2 (1.7) | 69.7 (2.0) | 69.5 (1.8) | 60.6 (0.5) | 72.1 (0.7) | **65.41** |
| + $TDL_{ToDD}$ | 70.1 (0.4) | 60.3 (0.3) | 59.1 (0.3) | 65.1 (1.4) | 71.4 (1.1) | 71.1 (0.7) | 64.9 (0.6) | 73.4 (0.8) | **66.93** |
| GraphLoG | 67.2 (0.2) | 57.9 (0.2) | 57.9 (0.3) | 57.8 (0.9) | 64.2 (1.1) | 65.0 (1.3) | 54.3 (0.7) | 72.3 (0.9) | 62.08 |
| + $TDL_{atom}$ | 72.1 (0.3) | 62.0 (0.2) | 60.7 (0.2) | 56.6 (0.8) | 73.0 (0.9) | 70.4 (0.9) | 61.2 (0.4) | 76.8 (0.7) | **66.59** |
| + $TDL_{ToDD}$ | 70.7 (0.2) | 60.7 (0.3) | 61.5 (0.3) | 59.5 (0.5) | 72.9 (1.8) | 71.6 (0.8) | 62.1 (0.3) | 80.1 (0.4) | **67.39** |

Table 3: Linear probing: given two molecules, predict distance between their PIs; MSE.

| | Tox21 | ToxCast | Sider | ClinTox | MUV | HIV | BBBP | Bace |
|---|---|---|---|---|---|---|---|---|
| ContextPred | 6.330 (0.007) | 5.351 (0.084) | 22.227 (0.102) | 4.234 (0.039) | 2.342 (0.001) | 6.127 (0.009) | 4.907 (0.072) | 3.591 (0.038) |
| + $TAE_{ahd}$ | 5.970 (0.002) | 5.263 (0.003) | 21.262 (0.070) | 4.255 (0.023) | 1.851 (0.003) | 5.752 (0.023) | 4.901 (0.024) | 3.069 (0.012) |
| GraphCL | 6.873 (0.011) | 5.440 (0.031) | 24.019 (0.059) | 4.537 (0.006) | 2.411 (0.006) | 5.996 (0.006) | 5.666 (0.003) | 3.858 (0.007) |
| + $TDL_{atom}$ | 5.371 (0.031) | 5.191 (0.023) | 20.641 (0.022) | 4.231 (0.025) | 1.645 (0.004) | 5.001 (0.022) | 4.936 (0.002) | 2.375 (0.008) |
| JOAO | 6.465 (0.032) | 5.277 (0.010) | 21.299 (0.085) | 4.277 (0.118) | 2.061 (0.001) | 5.418 (0.016) | 4.995 (0.036) | 2.782 (0.004) |
| + $TDL_{atom}$ | 5.642 (0.011) | 5.187 (0.010) | 20.515 (0.044) | 4.185 (0.027) | 1.749 (0.004) | 5.011 (0.028) | 4.979 (0.015) | 2.377 (0.002) |
| SimGRACE | 10.015 (0.050) | 13.387 (4.770) | 26.514 (0.049) | 4.669 (0.017) | 2.591 (0.005) | 6.150 (0.023) | 6.167 (0.035) | 13.054 (0.938) |
| + $TDL_{atom}$ | 5.359 (0.005) | 5.263 (0.016) | 20.630 (0.034) | 4.071 (0.042) | 1.545 (0.007) | 4.714 (0.033) | 5.131 (0.011) | 2.296 (0.003) |
| GraphLoG | 6.877 (0.012) | 5.279 (0.004) | 24.073 (0.038) | 4.482 (0.055) | 2.857 (0.004) | 6.591 (0.015) | 5.029 (0.079) | 3.918 (0.004) |
| + $TDL_{atom}$ | 5.814 (0.032) | 5.212 (0.018) | 21.441 (0.044) | 4.237 (0.015) | 1.719 (0.005) | 4.669 (0.010) | 4.962 (0.015) | 2.235 (0.005) |

the singular values of the covariance matrix of the representations, where vanishing values hint at collapsed dimensions. The phenomenon has recently been observed in computer vision and there are only initial explanations to date [Hua et al., 2021]. However, here, we observe the collapse only over the downstream data. Our hypothesis is that TDL's fine-grained supervision forces the models to capture relevant features, which they might have neglected otherwise. We also depict the graphs for SimGRACE and JOAO, which look very different, reflecting the variety of the approaches. There is basically no difference for JOAO(+TDL), while TDL shrinks the values for SimGRACE; the latter is less optimal and needs further investigation.

Table 4: Binary classification over MoleculeNet; ROC-AUC, % Pos. is min/med/max for multi-task.

| | Tox21 | ToxCast | Sider | ClinTox | MUV | HIV | BBBP | Bace | Average |
|---|---|---|---|---|---|---|---|---|---|
| # Molecules | 7,831 | 8,575 | 1,427 | 1,478 | 93,087 | 41,127 | 2,039 | 1,513 | - |
| # Tasks | 12 | 617 | 27 | 2 | 17 | 1 | 1 | 1 | |
| % Positives | 2.4/4.6/12.0 | 0.2/1.3/20.5 | 1.5/66.3/92.4 | 7.6/50.6/93.6 | 0.03/0.03/0.03 | 3.5 | 76.5 | 45.7 | |
| No pretrain (GIN) | 74.6 (0.4) | 61.7 (0.5) | 58.2 (1.7) | 58.4 (6.4) | 70.7 (1.8) | 75.5 (0.8) | 65.7 (3.3) | 72.4 (3.8) | 67.15 |
| AD-GCL [Suresh et al., 2021] | 76.5 (0.8) | 63.0 (0.7) | 63.2 (0.7) | 79.7 (3.5) | 72.3 (1.6) | 78.2 (0.9) | 70.0 (1.0) | 78.5 (0.8) | 72.67 |
| iMolCLR [Wang et al., 2022b] | 75.1 (0.7) | 63.5 (0.4) | 59.4 (1.0) | 81.0 (2.6) | 74.7 (1.9) | 77.3 (1.2) | 69.6 (1.2) | 77.3 (1.0) | 72.24 |
| Mole-BERT [Xia et al., 2023b] | 76.8 (0.5) | 64.3 (0.2) | 62.8 (1.1) | 78.9 (3.0) | 78.6 (1.8) | 78.2 (0.8) | 71.9 (1.6) | 80.8 (1.4) | 74.04 |
| SEGA [Wu et al., 2023] | 76.7 (0.4) | 65.2 (0.9) | 63.6 (0.3) | 84.9 (0.9) | 76.6 (2.4) | 77.6 (1.3) | 71.8 (1.0) | 77.0 (0.4) | 74.17 |
| TAE$_{ahd}$ | 75.2 (0.8) | 63.1 (0.3) | 61.9 (0.8) | 80.6 (1.9) | 74.6 (1.8) | 73.5 (2.1) | 67.5 (1.1) | 82.5 (1.1) | 72.36 |
| TAE$_{ToDD}$ | 76.8 (0.9) | 64.0 (0.5) | 61.9 (0.8) | 79.3 (3.6) | 75.8 (3.2) | 75.9 (1.1) | 70.4 (0.8) | 81.6 (1.4) | 73.22 |
| ContextPred | 75.7 (0.7) | 63.9 (0.6) | 60.9 (0.6) | 65.9 (3.8) | 75.8 (1.7) | 77.3 (1.0) | 68.0 (2.0) | 79.6 (1.2) | 70.89 |
| + TAE$_{ahd}$ | 76.4 (0.5) | 63.2 (0.4) | 62.0 (0.7) | 74.6 (4.4) | 76.7 (1.6) | 77.7 (1.2) | 68.9 (1.1) | 80.7 (1.6) | 72.53 |
| + TAE$_{ToDD}$ | 75.7 (0.4) | 63.1 (0.3) | 61.3 (0.5) | 72.1 (1.3) | 77.2 (1.8) | 77.6 (1.1) | 69.6 (0.9) | 80.1 (1.4) | 72.09 |
| GraphCL | 73.9 (0.7) | 62.4 (0.6) | 60.5 (0.9) | 76.0 (2.7) | 69.8 (2.7) | 78.5 (1.2) | 69.7 (0.7) | 75.4 (1.4) | 70.78 |
| + TDL$_{atom}$ | 75.3 (0.4) | 64.4 (0.3) | 61.2 (0.6) | 83.7 (2.7) | 75.7 (0.8) | 78.0 (0.9) | 70.9 (0.6) | 80.5 (0.8) | 73.71 |
| + TDL$_{ToDD}$ | 75.2 (0.7) | 64.2 (0.3) | 61.5 (0.4) | 85.2 (1.8) | 75.9 (2.1) | 77.9 (0.8) | 69.9 (0.9) | 81.2 (1.9) | 73.88 |
| JOAO | 75.0 (0.3) | 62.9 (0.5) | 60.0 (0.8) | 81.3 (2.5) | 71.7 (1.4) | 76.7 (1.2) | 70.2 (1.0) | 77.3 (0.5) | 71.89 |
| + TDL$_{atom}$ | 75.5 (0.3) | 63.8 (0.2) | 60.6 (0.5) | 76.8 (1.5) | 73.8 (1.9) | 78.3 (1.2) | 70.3 (0.5) | 78.7 (0.6) | 72.22 |
| + TDL$_{ToDD}$ | 75.2 (0.3) | 63.6 (0.2) | 61.6 (0.6) | 80.7 (3.3) | 74.6 (1.6) | 77.4 (0.9) | 71.3 (0.8) | 81.0 (2.2) | 73.18 |
| SimGRACE | 74.4 (0.3) | 62.6 (0.7) | 60.2 (0.9) | 75.5 (2.0) | 75.4 (1.3) | 75.0 (0.6) | 71.2 (1.1) | 74.9 (2.0) | 71.15 |
| + TDL$_{atom}$ | 74.7 (0.5) | 63.0 (0.3) | 59.5 (0.4) | 73.7 (1.5) | 75.9 (1.6) | 77.3 (1.1) | 69.5 (0.9) | 79.1 (0.5) | 71.59 |
| + TDL$_{ToDD}$ | 75.6 (0.4) | 63.3 (0.5) | 59.9 (0.8) | 82.4 (2.7) | 75.6 (2.0) | 76.1 (1.3) | 69.9 (0.8) | 78.9 (1.6) | 72.71 |
| GraphLoG | 75.0 (0.6) | 63.4 (0.6) | 59.3 (0.8) | 70.1 (4.6) | 75.5 (1.6) | 76.1 (0.8) | 69.6 (1.6) | 82.1 (1.0) | 71.43 |
| + TDL$_{atom}$ | 76.1 (0.7) | 63.7 (0.4) | 59.9 (1.0) | 75.7 (3.5) | 75.7 (1.2) | 76.2 (1.8) | 69.6 (1.2) | 82.2 (1.5) | 72.39 |
| + TDL$_{ToDD}$ | 75.9 (0.8) | 63.5 (0.7) | 63.4 (0.3) | 79.8 (1.9) | 75.6 (1.1) | 76.2 (1.6) | 70.7 (0.9) | 82.1 (1.9) | 73.39 |

**Linear Probing, Tables 2, 3, 10.** We evaluated extensively using a linear layer on the representations of the pre-trained graph encoders over the MoleculeNet data in terms of binary classification and a custom distance prediction task. The probing, also in terms of MLPs, demonstrates that PIs possess a complementary nature to ECFP. Furthermore, TAE, which we developed for comparison purposes only, is competitive with the baselines and improves ContextPred. In Appendix C, we additionally show some results over the more challenging activity cliff data. Apart from JOAO+TDL, where the results are mixed, TDL yields overall impressive increases. Interestingly, the more complex ToDD-based version is not always better than the simpler one.

**Discussion.** The results demonstrate that both TAE and TDL successfully approach the PI space in terms of distances between molecules. Yet, it also has to be noted that the PIs we chose do not always capture the main characteristics of the sample space, as seen for BBBP. Similarly, the mixed ROGI scores, which incorporate downstream labels, hint at a varied downstream performance. Nevertheless, the specifics of particular downstream datasets should not be confused with the more general molecular representation space. In that respect, especially TDL *has been proven effective overall in improving representation power in a variety of probing tasks for all baselines*. We have also shown that, on closer look, the effects can be very different with the considered baselines, depending on their individual deficiencies. For JOAO, we have seen an improvement in distance representation but not as large improvements thereafter, suggesting that it captures similar features already. Recall, however, that JOAO "only" adapts GraphCL in that it chooses between views more flexibly. In this regard, TDL represents a more efficient adaptation of GraphCL, which turns out to be also more effective in terms of SSL generally, as we show next.

## 4.2 Evaluation on Downstream Tasks

**Transfer Learning Benchmarks, Tables 4, 12, Figure 12.** First, note that the MoleculeNet benchmark contains various types of data. We have already observed the strong differences between Bace and BBBP in terms of our PIs. Here we additionally have some very large datsets and some with very many tasks; in both we expect the labels and their correlation to have increasing impact and reduce the one of the original embedding space. We report some most recent works that used the same pre-training data to provide a context for interpreting our results. As expected, TAE alone only reaches decent average performance, yet surpasses SOTA on ClinTox and Bace and also considerably improves ContextPred there. We see more variability than above in the effects TDL has on the individual baselines. We do not observe a general impact of dataset balance. To learn about the impact of labels the influence of multiple tasks, we ran some models also on individual tasks of the multi-task data and observe larger increases for TDL there (see figures in Appendix D). Altogether, the table shows mixed results for SimGRACE; some improvement for GraphLoG and, with a single exception, also for JOAO; and large improvements for GraphCL. Notably, when using the stronger

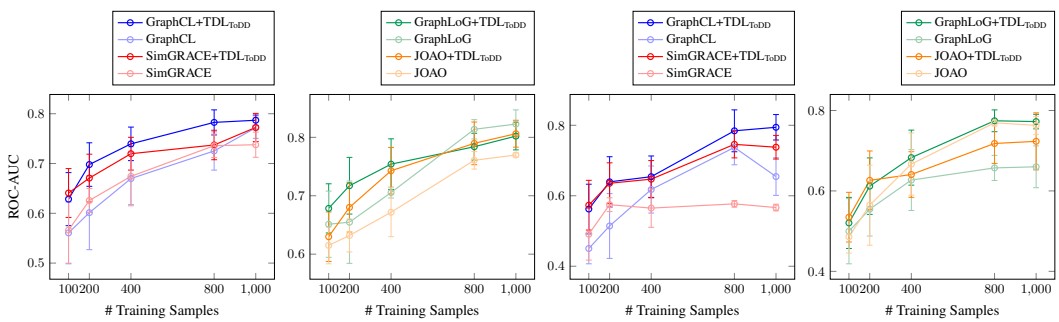

Figure 7: Performance over smaller datasets, subsets of Bace (left) and ClinTox (right).

ToDD filtration, TDL demonstrates convincing improvements across all baselines and makes them competitive with SOTA.

**The Low-Data Scenario, Figures 2, 7, 13, 14.** Inspired by the impressive increases we see in the quality of representations after pre-training, we experimented with considerably smaller datasets. This is a scenario which is relevant in practice since proprietary data from costly experiments is often only available in small numbers [Tom et al., 2023]. Here we see clearly that the improvement in representation space is reflected in downstream performance. There is still some dependence on the dataset but, overall, TDL often yields remarkable improvements. The introduction figure shows that also other models show room for improvement; note that GraphMVP uses 3D information.

**Further Analysis.** Due to space constraints, we present only the most impactful findings in this section and the rest in the appendix, in particular: (1) **Ablation Studies.** Our approach offers great variability in terms of, for instance, the graph encoder, the construction of the PIs, or even the type of topological fingerprint more generally. (2) **Unsupervised learning.** We observe similar but less pronounced trends, likely due to the smaller size of the data and fewer features available. (3) **PIs w/o SSL.** In view of the promising results with PIs in supervised ML [Krishnapriyan et al., 2021, Demir et al., 2022], we consider the PIs with XGB and SVM, our approaches based on pre-training are better. (4) **Other datasets.** We conducted experiments over several other datasets, including activity-cliff data, where TDL is shown beneficial more generally.

**Discussion.** In this section, we have shown results very different from the ones we usually see. In the benchmark, we observe some improvements and, for GraphCL+TDL, numbers competitive with SOTA, but they do not reach the general, large impact we saw in Section 4.1. However, we have such impact over smaller data. Altogether, TDL offers great benefits for *all* baselines we considered:

- It leads to **much better calibrated distances in the representation space** in terms of PH. Since the magnitude of increase is generally remarkable but varies with the data, this shows potential for further improvement, especially, in terms of choosing the right PIs.
- It considerably **improves representations** in terms of established SSL metrics.
- **Downstream performance reflects this gain of representation power specifically in the low-data scenario**. Given the importance of this kind of data in the real world and the deficiencies of the models we checked, we conclude that our proposal is not only novel and different in its distance-based modeling, but also that it advances molecular SSL in important aspects empirically.

## 5    Conclusions

In this paper, we focus on the quality of representations in molecular SSL, an important topic which has not been addressed greatly to date. We propose a novel approach of shaping the embedding space based on distances in terms of fingerprints from persistent homology. We have evaluated our topological-distance-based contrastive loss extensively, and show that it solves deficiencies of existing models and considerably improves various baselines in terms of representation power. In future work, we plan to analyze the relationship between specific PIs and downstream data in more detail, to further improve our approach, include external knowledge, and to put more focus on regression tasks.

## Acknowledgments

This work was supported by National Key R&D Program of China (2021YFB3500700), NSFC Grant 62172026, National Social Science Fund of China 22&ZD153, the Fundamental Research Funds for the Central Universities and SKLSDE; correspondence to Lei Shi.

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

# A  Datasets and Experimental Details

**Pre-training Datasets.** As usual, for GNN pre-training, we use a minimal set of node and bond features that unambiguously describe the two-dimensional structure of molecules from the ZINC15 database following previous works [Hu* et al., 2020].

- Node features:
    - Atom number: $[1, 118]$
    - Chirality tag: {unspecified, tetrahedral cw, tetrahedral ccw, other}
- Edge features:
    - Bond type: {single, double, triple, aromatic}
    - Bond direction: {−, endupright, enddownright}

We note that the atom number implicitly provides domain knowledge in that it captures important knowledge about the atom's charge.

**Downstream Task Datasets.** We focus on molecular property prediction, where we adopt the widely-used 8 binary classification datasets contained in MoleculeNet [Wu et al., 2018a]. In addition, we experimented with 4 regression tasks from various low-data domains in line with the same setting as [Liu et al., 2022a]. We provide detailed information of these datasets in Table 5 and more information can be found in [Liu et al., 2022a]. To further evaluate the usefulness of our models, we also use the opioids-related datasets from [Deng et al., 2022]. The number (percentage) of activity cliff (AC) molecules are summarized in Table 6. Notably, among MDR1, MOR, DOR and KOR, nearly half molecules are equipped with activity cliff scaffolds. Finally, scaffold-split [Ramsundar et al., 2019] is used to splits graphs into train/val/test set as 80%/10%/10% which mimics real-world use cases.

Table 5: Summary for the molecule datasets.

| Dataset | Task | # Tasks | # Molecules |
|---|---|---|---|
| BBBP | Classification | 1 | 2,039 |
| Tox21 | Classification | 12 | 7,831 |
| ToxCast | Classification | 617 | 8,576 |
| Sider | Classification | 27 | 1,427 |
| ClinTox | Classification | 2 | 1,478 |
| MUV | Classification | 17 | 93,087 |
| HIV | Classification | 1 | 41,127 |
| Bace | Classification | 1 | 1,513 |
| ESOL | Regression | 1 | 1,128 |
| Lipo | Regression | 1 | 4,200 |
| Malaria | Regression | 1 | 9,999 |
| CEP | Regression | 1 | 29,978 |

Table 6: Summary of activity cliffs in the opioids datasets.

| Dataset | MDR1 | CYP2D6 | CYP3A4 | MOR | DOR | KOR |
|---|---|---|---|---|---|---|
| # Mol. AC. Scaff. (%) | 594 (41.3) | 710 (31.0) | 926 (25.2) | 1,627 (46.1) | 1,342 (41.6) | 1,502 (45.2) |
| Task | Classification | Classification | Classification | Classification | Classification | Classification |
| # Tasks | 1 | 1 | 1 | 1 | 1 | 1 |

**Computing Environment.** The experiments are conducted with two RTX 3090 GPUs.

**Model Configurations.** Following [Hu* et al., 2020], we adopt a 5-layer Graph Isomorphism Network (GIN) [Xu et al.] with 300-dimensional hidden units as the backbone architecture. We use mean pooling as the readout function. During the pre-training stage, GNNs are pre-trained for 100 epochs with batch-size as 256 and the learning rate as 0.001. During the fine-tuning stage, we train for 100 epochs with batch-size as 32, dropout rate as 0.5, and report the test performance using ROC-AUC at the best validation epoch. To evaluate the learned representations, we follow the linear probing (linear evaluation) [Akhondzadeh et al., 2023], where a linear classifier (1 linear layer) is

trained on top of the frozen backbone. We probe two types of properties from the representation: the first is the molecular properties of downstream tasks, for which we use the same settings as fine-tuning and report ROC-AUC; the second is topological fingerprints (PIs) of downstream molecular graphs, for which we train for 50 epochs with batch-size as 256, dropout rate as 0.0, and report MSE and Pearson correlation coefficients between distances in embedding space and corresponding PI space. The experiments are run with 10 different random seeds, and we report mean and standard deviation.

**Performance Comparison.** We compare the proposed method with existing self-supervised graph representation learning algorithms (i.e. InfoGraph [Sun et al., 2019], EdgePred [Hamilton et al., 2017], ContextPred [Hu* et al., 2020], AttrMask [Hu* et al., 2020], GraphLoG [Xu et al., 2021], iMolCLR [Wang et al., 2022b], AD-GCL [Suresh et al., 2021], JOAO [You et al., 2021], SimGRACE [Xia et al., 2022], GraphCL [You et al., 2020], GraphMAE [Hou et al., 2022], MGSSL [Zhang et al., 2021], GPT-GNN [Hu et al., 2020], G-Contextual [Rong et al., 2020], G-Motif [Rong et al., 2020], GraphMVP [Liu et al., 2022a]). We report the results from their own papers with two exceptions: (1) for GraphMAE and GraphLoG, they reported the test performance of the last epoch; and (2) MGSSL and SimGRACE, they only used 3 random seeds; and (3) iMolCLR used ~10M molecules for pre-training. We reproduce the results of these four models with the same protocol as [Hu* et al., 2020], utilizing the pre-trained models originally provided by the respective papers.

**Topological Fingerprints.** We set the highest dimension in the simplicial complex to 1. This doesn't impact performance because molecular graphs, being planar, rarely possess loops of length 3; in fact, the majority of loops exhibit a length of 5 or more.

Firstly, we use the sublevel filtration method. In terms of filtration functions, we use atomic number. For every molecular graph, we utilize an extended persistence module [Cohen-Steiner et al., 2009, Yan et al., 2022] to compute persistent diagrams (as depicted in Figure 1 of [Yan et al., 2022]) and then vectorize them to get PIs as topological fingerprints. The extended module incorporates a additional reverse process. Through this module, all loop features created will be killed in the end. As an example, in Figure 1, a 1-dim extended persistence point that can be captured is $(t_4, t_2)$. This is because the loop feature is born at time $t_4$ and then dies at time $t_2$ in the reverse process. Note that we only use atomic number as filtration functions for TDL$_{atom}$. In addition to the atomic number, we also investigate the heat kernel signature (hks) with temperature values $t = 0.1$ [Sun et al., 2009] and node degree as filtration functions. Observe that this is more geometric information, compared to the domain knowledge encoded in the atomic number (atom). Considering that TAE is a straightforward model and there is too little information for PI$_{atom}$, we concatenated these 3 PIs as reconstruction targets for TAE$_{ahd}$.

We have also considered a specific filtration function from ToDD [Demir et al., 2022], which incorporates atomic mass and, additionally, partial charges and bond types inside a so-called multi-VR filtration. In a nutshell, for each of those three types of information, we do not simply consider the given filtration, but construct a 2D filtration by filtering: in one dimension according to the above information and in the second dimension according to a VR filtration capturing the distances between atoms (Figure 5 in [Demir et al., 2022] illustrates this kind of filtration well.) Lastly, the 3 2D PIs are concatenated, and we compute distances based on the combination of PIs. In Appendix G, we report ablation results for various filtrations.

In order to explore the effectiveness of PIs, we calculate the widely-used Tanimoto coefficient [Bajusz et al., 2015] of the extended connectivity fingerprints (ECFPs) [Rogers and Hahn, 2010] between two molecules as their chemical similarity on 8 downstream tasks datasets. Then, we pick the molecule pairs with top 20% similarity as 'similar' ones (Blue) and the left 80% molecule pairs in the datasets are 'random' pairs (Yellow). In final, we calculate the cosine similarity of these molecule pairs based on PIs. Figure 8 shows molecules with similar ECFPs have rather similar PIs.

It is computationally efficient to apply topological fingerprints method. Extracting and computing PI$_{atom}$ for 2M molecular graphs from ZINC15 dataset only spent 20 minutes in a single workstation with 8-core CPU, 64 GB of memory.

**TAE+ Models.** We hypothesize that TAE gives other pre-training approach a good initial embedding space to start from. Therefore, this pre-training strategy is to first perform TAE pre-training and then other models pre-training. We study TAE+EdgePred, TAE+AttrMask and TAE+ContextPred.

**TDL.** The temperature parameter $\tau$ is set to 0.1.

**TDL over Views.** Like any other contrastive loss function, TDL can be utilized as a standalone objective by considering views. Here, we show preliminary experiments using a formulation based on the loss of GraphCL [You et al., 2020], we also use its augmentations. A minibatch of $N$ graphs is processed through contrastive learning, resulting in $2N$ augmented graphs. We denote the augmented views for the $n$th graph in the minibatch as $z_{n,i}$ and $z_{n,j}$, and modify TDL as follows:

$$\mathcal{L}_{\text{TDL}_n}^{views} = \frac{1}{N-1} \sum_{m \in [1,N]} - \log \frac{\exp\left(sim\left(z_{n,i}, z_{m,j}\right)/\tau\right)}{\sum_{k \in [1,N]} \mathbb{I}_{[dis(I_{n,i}, I_{k,j}) \geq dis(I_{n,i}, I_{m,j})]} \cdot \exp\left(sim\left(z_{n,i}, z_{k,j}\right)/\tau\right)}$$

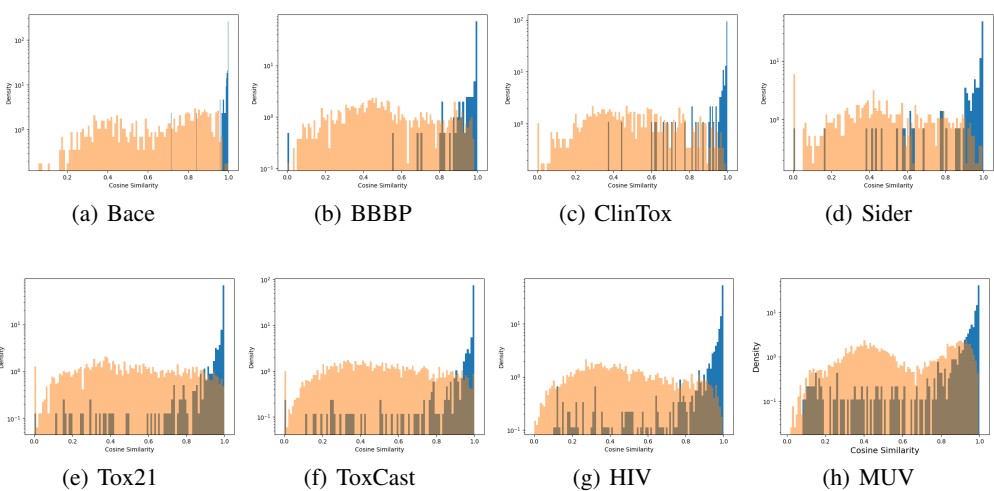

(a) Bace     (b) BBBP     (c) ClinTox     (d) Sider

(e) Tox21     (f) ToxCast     (g) HIV     (h) MUV

Figure 8: Similarity histograms of PIs on 8 downstream datasets. Cosine similarity measures the similarity between the PIs while 'Random' (Yellow) and 'Similar' (Blue) are defined by the similarity between ECFPs.

# B   Analysis of Embeddings

Table 7: The Pearson correlation coefficient between real PIs and reconstructed PIs (i.e., TAE's embeddings) on the downstream data (the full dataset), after pre-training. We also report the overlap of the downstream data with the pre-training data, but do not observe considerable impact here.

|  | Tox21 | ToxCast | Sider | ClinTox | MUV | HIV | BBBP | Bace |
|---|---|---|---|---|---|---|---|---|
| # Molecules | 7,831 | 8,575 | 1,427 | 1,478 | 93,087 | 41,127 | 2,039 | 1,513 |
| # Molecules in ZINC15 | 628 (8%) | 608 (7%) | 1 (0%) | 51 (4%) | 7599 (8%) | 925 (2%) | 100 (5%) | 0 (0%) |
| TAE$_{\text{ahd}}$ | 0.8572 | 0.7744 | 0.5939 | 0.8642 | 0.9044 | 0.7359 | 0.8660 | 0.8514 |

Table 8: 5-nearest neighbors classifier of CL methods for 8 downstream datasets, ROC-AUC (%).

| | Tox21 | ToxCast | Sider | ClinTox | MUV | HIV | BBBP | Bace | Average |
|---|---|---|---|---|---|---|---|---|---|
| $PI_{atom}$ | 58.8 | 51.1 | 58.7 | 50.9 | 50.2 | 64.8 | 55.2 | 75.5 | 58.15 |
| $PI_{ToDD}$ | 62.7 | 51.7 | 56.3 | 64.9 | 49.7 | 63.9 | 56.7 | 68.7 | 59.33 |
| ECFP | 63.8 | 54.6 | 59.1 | 50.7 | 54.0 | 68.1 | 59.3 | 77.0 | 60.82 |
| GraphCL | 65.3 | 55.8 | 60.6 | 56.4 | 52.0 | 70.4 | 57.9 | 71.4 | 61.23 |
| GraphCL + $TDL_{atom}$ | 67.2 | 56.7 | 62.6 | 57.5 | 54.4 | 71.1 | 62.0 | 70.7 | **62.78** |
| GraphCL + $TDL_{ToDD}$ | 68.0 | 56.5 | 61.6 | 63.5 | 54.1 | 70.3 | 62.5 | 66.9 | **62.93** |
| JOAO | 66.3 | 55.7 | 59.9 | 57.2 | 53.2 | 72.0 | 58.9 | 70.2 | 61.67 |
| JOAO + $TDL_{atom}$ | 66.9 | 55.4 | 58.9 | 62.5 | 52.7 | 69.1 | 60.1 | 70.4 | **62.00** |
| JOAO + $TDL_{ToDD}$ | 67.1 | 56.8 | 60.5 | 61.2 | 51.5 | 67.7 | 59.5 | 72.7 | **62.13** |
| SimGRACE | 64.2 | 56.1 | 58.3 | 46.7 | 55.1 | 68.2 | 58.4 | 70.4 | 59.67 |
| SimGRACE + $TDL_{atom}$ | 66.1 | 56.3 | 61.9 | 68.4 | 52.4 | 71.3 | 63.6 | 77.3 | **64.66** |
| SimGRACE + $TDL_{ToDD}$ | 67.3 | 58.1 | 60.2 | 77.5 | 52.2 | 68.3 | 67.8 | 76.7 | **66.01** |
| GraphLoG | 63.0 | 55.9 | 59.8 | 68.9 | 52.2 | 68.2 | 56.8 | 79.4 | 63.02 |
| GraphLoG + $TDL_{atom}$ | 66.7 | 57.3 | 60.9 | 67.3 | 50.0 | 68.4 | 63.6 | 77.5 | **63.96** |
| GraphLoG + $TDL_{ToDD}$ | 67.4 | 56.9 | 61.7 | 58.2 | 53.1 | 72.1 | 63.8 | 78.2 | **63.92** |

Table 9: ROGI scores, lower score means smoother embedding space in terms of downstream labels.

| | ClinTox | Bace | BBBP | Sider |
|---|---|---|---|---|
| GraphCL | 0.4496 | 0.7883 | 0.7033 | 0.6863 |
| GraphCL + $TDL_{atom}$ | 0.4549 | 0.7623 | 0.6973 | 0.7213 |
| JOAO | 0.4675 | 0.7900 | 0.7344 | 0.7320 |
| JOAO + $TDL_{atom}$ | 0.4668 | 0.7673 | 0.7284 | 0.7284 |
| SimGRACE | 0.4722 | 0.7919 | 0.7874 | 0.7308 |
| SimGRACE + $TDL_{atom}$ | 0.4594 | 0.7635 | 0.6989 | 0.7004 |
| GraphLoG | 0.4456 | 0.7480 | 0.6908 | 0.6815 |
| GraphLoG + $TDL_{atom}$ | 0.4789 | 0.8161 | 0.7167 | 0.7320 |

**Alignment Analysis.** To evaluate alignment, we construct positive and negative molecule pairs. Positive pairs in a dataset are defined as those sharing identical molecular properties, while negative pairs exhibit differing properties. We randomly select 10k positive and negative pairs of molecular graphs from the Bace, BBBP, Clintox, and Sider datasets. Subsequently, we compute the normalized Euclidean distance of the embeddings and illustrate the results via a histogram, as depicted in Figure 9 and Figure 10. We apply TDL on top of GraphCL, SimGRACE, JOAO, and GraphLoG to represent the impact of TDL on the baselines.

The normalized curves demonstrate significant differences in the embedding space among different models in the distance dimension. It is observable that TDL refines the distribution of distances to more closely align with that of PIs. The distinction between $TDL_{ToDD}$ and $TDL_{atom}$ explains why $TDL_{atom}$ does not enhance JOAO because JOAO's curve shows that it is in fact possible to get such an atom PIs' distribution. Furthermore, JOAO reveals that additional domain knowledge beyond atom can be effectively incorporated via $TDL_{ToDD}$.

## C  Additional Linear Probing

We continue with linear probing for molecular properties using the more challenging activity cliff dataset. In Table 10, we show that TDL often yields improvements.

## D  Additional Benchmark Results

**Primary Results.** The main results on 8 molecular property prediction tasks are listed in Table 11. Note that all SSL methods in Table 11 are pre-trained on the same dataset based on ZINC15. We observe that *the average performance of GraphCL+TDL has surpassed these existing SSL methods.*

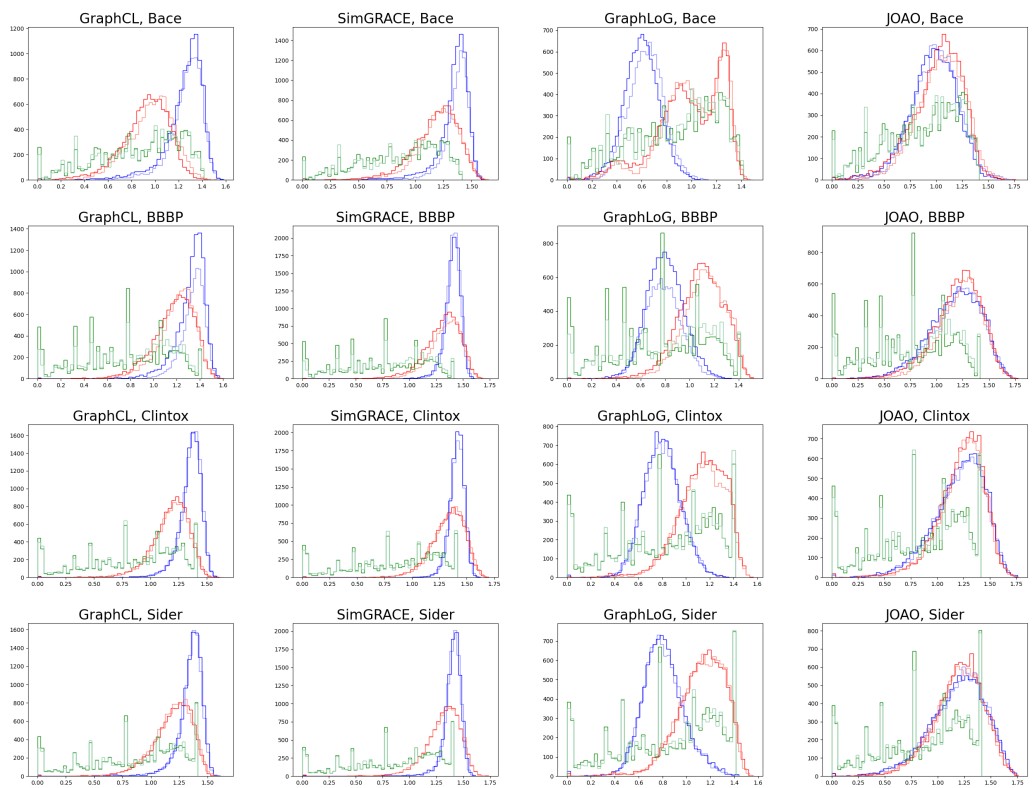

Figure 9: Evaluating alignment in the embedding space influenced by $TDL_{atom}$. Dark/light shades denote positive/negative pairs, respectively. Blue illustrates the embedding of baselines, red depicts the embedding of baselines+$TDL_{atom}$, and green represents $PI_{atom}$.

Table 10: Linear evaluation of molecular property prediction using opioids-related datasets in classification task, repeated 30 times with 30 different seeds, ROC-AUC (%).

| Dataset | MDR1 | | CYP2D6 | | CYP3A4 | |
|---|---|---|---|---|---|---|
| | ROC ↑ | PRC ↑ | ROC ↑ | PRC ↑ | ROC ↑ | PRC ↑ |
| GraphCL | 0.832 (0.172) | 0.674 (0.256) | 0.512 (0.205) | 0.019 (0.017) | 0.763 (0.147) | 0.327 (0.258) |
| GraphCL + $TDL_{atom}$ | 0.928 (0.081) | 0.736 (0.241) | 0.590 (0.251) | 0.032 (0.043) | 0.817 (0.133) | 0.296 (0.183) |

| Dataset | MOR | | DOR | | KOR | |
|---|---|---|---|---|---|---|
| | ROC ↑ | PRC ↑ | ROC ↑ | PRC ↑ | ROC ↑ | PRC ↑ |
| GraphCL | 0.789 (0.060) | 0.595 (0.108) | 0.730 (0.065) | 0.382 (0.091) | 0.801 (0.048) | 0.593 (0.099) |
| GraphCL + $TDL_{atom}$ | 0.797 (0.050) | 0.601 (0.109) | 0.760 (0.065) | 0.432 (0.117) | 0.802 (0.063) | 0.598 (0.105) |

Additionally, we also apply TDL on both AD-GCL and iMolCLR, with results in the table showing that TDL still provides good improvements.

**Individual Tasks.** In Table 4, we observe that SimGRACE+TDL performs poorly on Sider and ClinTox. To learn about the influence of multiple tasks, we run SimGRACE and GraphLoG on individual tasks of Sider and ClinTox. Figure 12 shows increases for $TDL_{atom}$. We hence conclude that the task correlation is exploited by the models considerably during fine-tuning in the multi-task setting, such that the improved SSL representation (which likely does not fit every downstream task under consideration) provided by TDL has less impact.

**Activity Cliff Dataset.** We also evaluate our methods using the opioids-related datasets. We fine-tune GraphCL and GraphCL+TDL and observe that GraphCL+TDL outperformes GraphCL, as shown in the Table 12. This result is interesting in itself since it is often argued that SSL based on similarity in "just" the geometric space (i.e., ignoring domain knowledge about, e.g., matching pairs or activity cliffs) may yield problems in practice. However, here, we observe that increased power of SSL

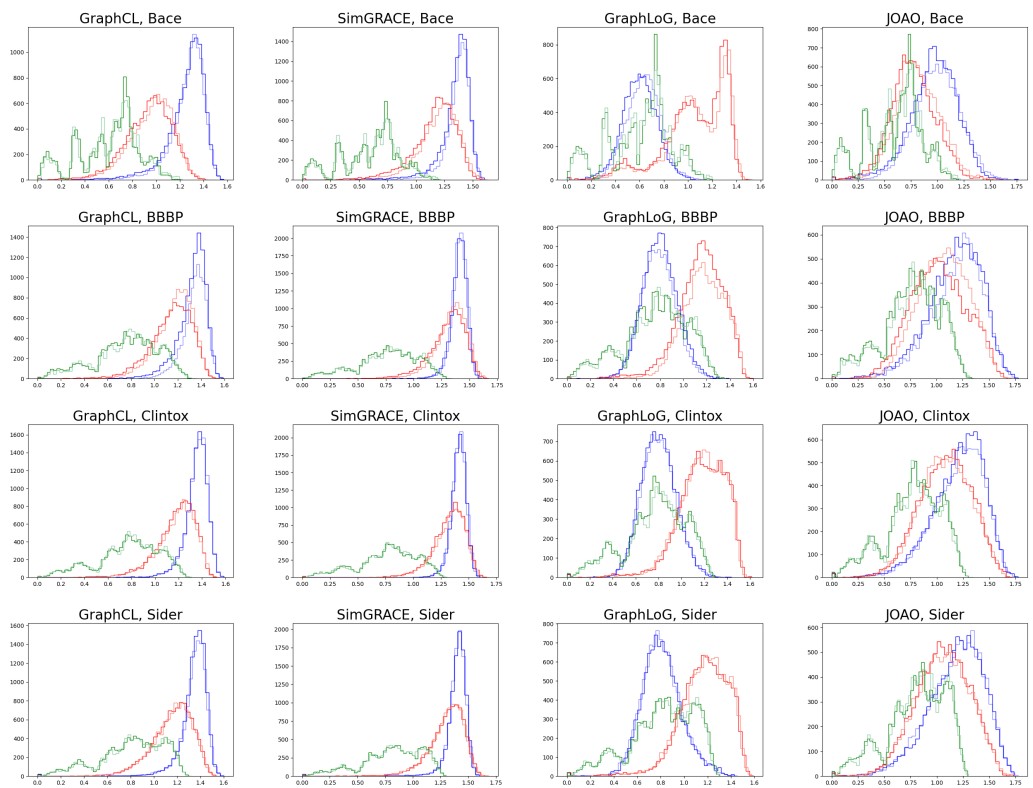

Figure 10: Evaluating alignment in the embedding space influenced by TDL$_{\text{ToDD}}$. Dark/light shades denote positive/negative pairs, respectively. Blue illustrates the embedding of baselines, red depicts the embedding of baselines+TDL$_{\text{ToDD}}$, and green represents PI$_{\text{ToDD}}$.

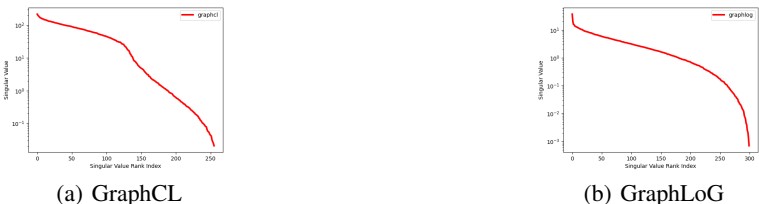

(a) GraphCL                    (b) GraphLoG

Figure 11: Singular values of covariance matrices of the representations; Pre-training datasets.

representation helps to increase the performance of existing methods, even on AC data - likely also because they have issues that still need to be resolved, such as the dimensional collapse we observed.

**Additional Pre-training Data.** Additionally, we validate the effectiveness of our methods on another pre-training dataset, GEOM (50K) [Liu et al., 2022a]. For an evaluation, we study TAE and apply TDL on GraphCL and JOAO. We follow all pre-training and fine-tuning settings of [Liu et al., 2022a], with the only difference being that we pre-train our models for 200 epochs instead. We compare the results from their original paper. As shown in Table 13, TAE alone reaches decent average performance, and surpasses SOTA on ClinTox and HIV. And we report the performance in regressive property prediction in Table 14 which shows that TDL enhances GraphCL and JOAO on many datasets. Please note that we considered these experiments only to obtain initial estimates; the GEOM data (which we used to be able to compare to related works) itself is too small for real pre-training if the basleline models are not able to exploit the 3D nature of the conformers.

Table 11: Results for 8 molecular property prediction tasks on ZINC15. For each downstream task, we report the mean (and standard deviation) ROC-AUC of 10 seeds.

| | Tox21 | ToxCast | Sider | ClinTox | MUV | HIV | BBBP | Bace | Average |
|---|---|---|---|---|---|---|---|---|---|
| # Molecules | 7,831 | 8,575 | 1,427 | 1,478 | 93,087 | 41,127 | 2,039 | 1,513 | - |
| No pretrain | 74.6 (0.4) | 61.7 (0.5) | 58.2 (1.7) | 58.4 (6.4) | 70.7 (1.8) | 75.5 (0.8) | 65.7 (3.3) | 72.4 (3.8) | 67.15 |
| InfoGraph | 73.3 (0.6) | 61.8 (0.4) | 58.7 (0.6) | 75.4 (4.3) | 74.4 (1.8) | 74.2 (0.9) | 68.7 (0.6) | 74.3 (2.6) | 70.10 |
| GraphLoG | 75.0 (0.6) | 63.4 (0.6) | 59.3 (0.8) | 70.1 (4.6) | 75.5 (1.6) | 76.1 (0.8) | 69.6 (1.6) | 82.1 (1.0) | 71.43 |
| JOAO | 75.0 (0.3) | 62.9 (0.5) | 60.0 (0.8) | 81.3 (2.5) | 71.7 (1.4) | 76.7 (1.2) | 70.2 (1.0) | 77.3 (0.5) | 71.89 |
| SimGRACE | 74.4 (0.3) | 62.6 (0.7) | 60.2 (0.9) | 75.5 (2.0) | 75.4 (1.3) | 75.0 (0.6) | 71.2 (1.1) | 74.9 (2.0) | 71.15 |
| GraphMAE | 75.2 (0.9) | 63.6 (0.3) | 60.5 (1.2) | 76.5 (3.0) | 76.4 (2.0) | 76.8 (0.6) | 71.2 (1.0) | 78.2 (1.5) | 72.30 |
| MGSSL | 75.2 (0.6) | 63.3 (0.5) | 61.6 (1.0) | 77.1 (4.5) | 77.6 (0.4) | 75.8 (0.4) | 68.8 (0.6) | 78.8 (0.9) | 72.28 |
| TDL$_{atom}$ | 75.8 (0.5) | 62.1 (0.5) | 62.2 (0.9) | 79.1 (3.8) | 75.2 (2.3) | 76.9 (0.9) | 66.5 (1.8) | 78.4 (1.1) | 72.02 |
| TDL$_{atom}^{views}$ | 75.7 (0.3) | 64.3 (0.5) | 61.5 (0.5) | 80.7 (2.9) | 73.3 (1.4) | 77.6 (0.8) | 70.5 (0.9) | 80.6 (1.1) | 73.02 |
| EdgePred | 76.0 (0.6) | 64.1 (0.6) | 60.4 (0.7) | 64.1 (3.7) | 74.1 (2.1) | 76.3 (1.0) | 67.3 (2.4) | 79.9 (0.9) | 70.27 |
| TAE$_{ahd}$ + EdgePred | 77.2 (0.4) | 63.8 (0.5) | 60.1 (0.7) | 72.2 (2.4) | 75.9 (1.7) | 76.8 (1.0) | 68.1 (0.8) | 81.5 (1.4) | **71.95** |
| AttrMask | 76.7 (0.4) | 64.2 (0.5) | 61.0 (0.7) | 71.8 (4.1) | 74.7 (1.4) | 77.2 (1.1) | 64.3 (2.8) | 79.3 (1.6) | 71.15 |
| TAE$_{ahd}$ + AttrMask | 76.6 (0.2) | 64.0 (0.3) | 58.2 (1.3) | 71.6 (3.7) | 75.5 (2.1) | 77.2 (0.6) | 68.7 (0.7) | 81.0 (1.1) | **71.60** |
| ContextPred | 75.7 (0.7) | 63.9 (0.6) | 60.9 (0.6) | 65.9 (3.8) | 75.8 (1.7) | 77.3 (1.0) | 68.0 (2.0) | 79.6 (1.2) | 70.89 |
| TAE$_{ahd}$ + ContextPred | 76.4 (0.5) | 63.2 (0.4) | 62.0 (0.7) | 74.6 (4.4) | 76.7 (1.6) | 77.7 (1.2) | 68.9 (1.1) | 80.7 (1.6) | **72.53** |
| GraphCL | 73.9 (0.7) | 62.4 (0.6) | 60.5 (0.9) | 76.0 (2.7) | 69.8 (2.7) | 78.5 (1.2) | 69.7 (0.7) | 75.4 (1.4) | 70.78 |
| GraphCL + TDL$_{atom}$ | 75.3 (0.4) | 64.4 (0.3) | 61.2 (0.6) | 83.7 (2.7) | 75.7 (0.8) | 78.0 (0.9) | 70.9 (0.6) | 80.5 (0.8) | **73.71** |
| AD-GCL | 76.5 (0.8) | 63.0 (0.7) | 63.2 (0.7) | 79.7 (3.5) | 72.3 (1.6) | 78.2 (0.9) | 70.0 (1.0) | 78.5 (0.8) | 72.67 |
| AD-GCL + TDL$_{atom}$ | 75.1 (0.5) | 64.5 (0.6) | 60.9 (1.3) | 79.4 (2.8) | 74.4 (1.7) | 77.4 (0.9) | 71.9 (1.1) | 82.1 (1.4) | **73.21** |
| iMolCLR | 75.1 (0.7) | 63.5 (0.4) | 59.4 (1.0) | 74.7 (1.9) | 81.0 (2.6) | 77.3 (1.2) | 69.6 (1.2) | 77.3 (1.0) | 72.24 |
| iMolCLR + TDL$_{atom}$ | 75.9 (0.6) | 63.7 (0.3) | 60.7 (0.8) | 83.8 (1.9) | 75.1 (1.3) | 76.7 (0.7) | 71.2 (0.9) | 78.5 (1.3) | **73.20** |

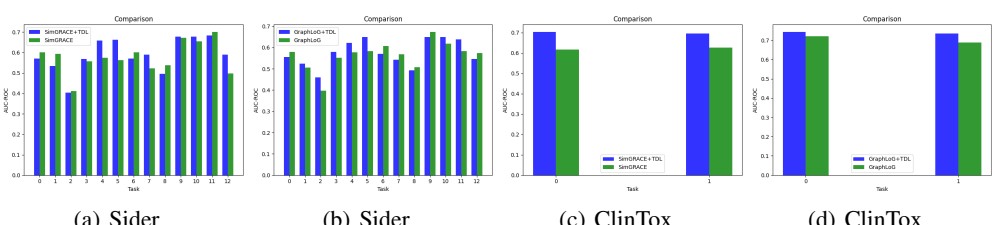

| (a) Sider | (b) Sider | (c) ClinTox | (d) ClinTox |
|---|---|---|---|

Figure 12: Performance on individual tasks of the multi-task data. We report the mean ROC-AUC of 10 seeds.

Table 12: Results for molecular property prediction using opioids-related datasets in classification tasks, repeated 30 times with 30 different seeds, ROC-AUC (%).

| Dataset | MDR1 | | CYP2D6 | | CYP3A4 | |
|---|---|---|---|---|---|---|
| | ROC ↑ | PRC ↑ | ROC ↑ | PRC ↑ | ROC ↑ | PRC ↑ |
| GraphCL | 0.912 (0.152) | 0.766 (0.211) | 0.558 (0.216) | 0.079 (0.143) | 0.803 (0.155) | 0.310 (0.224) |
| GraphCL + TDL$_{atom}$ | 0.934 (0.126) | 0.770 (0.233) | 0.565 (0.261) | 0.082 (0.116) | 0.815 (0.159) | 0.418 (0.245) |

| Dataset | MOR | | DOR | | KOR | |
|---|---|---|---|---|---|---|
| | ROC ↑ | PRC ↑ | ROC ↑ | PRC ↑ | ROC ↑ | PRC ↑ |
| GraphCL | 0.861 (0.040) | 0.738 (0.105) | 0.804 (0.038) | 0.532 (0.076) | 0.859 (0.044) | 0.701 (0.085) |
| GraphCL + TDL$_{atom}$ | 0.879 (0.036) | 0.741 (0.093) | 0.821 (0.049) | 0.566 (0.092) | 0.861 (0.041) | 0.718 (0.058) |

Table 13: Results for 8 molecular property prediction tasks on GEOM. For each downstream task, we report the mean (and standard deviation) ROC-AUC of 3 seeds with scaffold splitting. Note that GraphMVP uses 3D geometry (conformers), while other models do not.

| | Tox21 | ToxCast | Sider | ClinTox | MUV | HIV | BBBP | Bace | Average |
|---|---|---|---|---|---|---|---|---|---|
| # Molecules | 7,831 | 8,575 | 1,427 | 1,478 | 93,087 | 41,127 | 2,039 | 1,513 | - |
| No pretrain | 74.6 (0.4) | 61.7 (0.5) | 58.2 (1.7) | 58.4 (6.4) | 70.7 (1.8) | 75.5 (0.8) | 65.7 (3.3) | 72.4 (3.8) | 67.15 |
| InfoGraph | 73.0 (0.7) | 62.0 (0.3) | 59.2 (0.2) | 75.1 (5.0) | 74.0 (1.5) | 74.5 (1.8) | 69.2 (0.8) | 73.9 (2.5) | 70.10 |
| EdgePred | 74.5 (0.4) | 60.8 (0.5) | 56.7 (0.1) | 55.8 (6.2) | 73.3 (1.6) | 75.1 (0.8) | 64.5 (3.1) | 64.6 (4.7) | 65.64 |
| ContextPred | 73.3 (0.5) | 62.8 (0.3) | 59.3 (1.4) | 73.7 (4.0) | 72.5 (2.2) | 75.8 (1.1) | 71.2 (0.9) | 78.6 (1.4) | 70.89 |
| AttrMask | 74.2 (0.8) | 62.5 (0.4) | 60.4 (0.6) | 68.6 (9.6) | 73.9 (1.3) | 74.3 (1.3) | 70.2 (0.5) | 77.2 (1.4) | 70.16 |
| GPT-GNN | 75.3 (0.5) | 62.2 (0.1) | 57.5 (4.2) | 57.8 (3.1) | 76.1 (2.3) | 75.1 (0.2) | 64.5 (1.1) | 77.6 (0.5) | 68.27 |
| GraphLoG | 73.0 (0.3) | 62.2 (0.4) | 57.4 (2.3) | 62.0 (1.8) | 73.1 (1.7) | 73.4 (0.6) | 67.8 (1.7) | 78.8 (0.7) | 68.47 |
| G-Contextual | 75.2 (0.3) | 62.6 (0.3) | 58.4 (0.6) | 59.9 (8.2) | 72.3 (0.9) | 75.9 (0.9) | 70.3 (1.6) | 79.2 (0.3) | 69.21 |
| G-Motif | 73.2 (0.8) | 62.6 (0.5) | 60.6 (1.1) | 77.8 (2.0) | 73.3 (2.0) | 73.8 (1.4) | 66.4 (3.4) | 73.4 (4.0) | 70.14 |
| GraphCL | 75.0 (0.3) | 62.8 (0.2) | 60.1 (1.3) | 78.9 (4.2) | 77.1 (1.0) | 75.0 (0.4) | 67.5 (3.3) | 68.7 (7.8) | 70.64 |
| JOAO | 74.4 (0.7) | 62.7 (0.6) | 60.7 (1.0) | 66.3 (3.9) | 77.0 (2.2) | 76.6 (0.5) | 66.0 (0.6) | 72.9 (2.0) | 69.57 |
| GraphMVP | 74.5 (0.4) | 62.7 (0.1) | 62.3 (1.6) | 79.0 (2.5) | 75.0 (1.4) | 74.8 (1.4) | 68.5 (0.2) | 76.8 (1.1) | 71.69 |
| GraphMVP-C | 74.4 (0.2) | 63.1 (0.4) | 63.9 (1.2) | 77.5 (4.2) | 75.0 (1.0) | 77.0 (1.2) | 72.4 (1.6) | 81.2 (0.9) | 73.07 |
| TAE$_{ahd}$ | 74.9 (0.3) | 62.3 (0.3) | 60.2 (0.9) | 81.9 (2.5) | 71.2 (1.4) | 77.6 (1.2) | 70.9 (3.3) | 78.4 (1.6) | 72.18 |

Table 14: Results for 4 molecular property prediction tasks (regression) on GEOM. For each downstream task, we report the mean (and standard variance) RMSE of 3 seeds with scaffold splitting.

| | ESOL | Lipo | Malaria | CEP | Average |
|---|---|---|---|---|---|
| No pretrain | 1.178 (0.044) | 0.744 (0.007) | 1.127 (0.003) | 1.254 (0.030) | 1.07559 |
| AttrMask | 1.112 (0.048) | 0.730 (0.004) | 1.119 (0.014) | 1.256 (0.000) | 1.05419 |
| ContextPred | 1.196 (0.037) | 0.702 (0.020) | 1.101 (0.015) | 1.243 (0.025) | 1.06059 |
| GraphMVP | 1.091 (0.021) | 0.718 (0.016) | 1.114 (0.013) | 1.236 (0.023) | 1.03968 |
| TAE$_{ahd}$ | 1.129 (0.019) | 0.723 (0.006) | 1.105 (0.014) | 1.265 (0.018) | 1.05552 |
| JOAO | 1.120 (0.019) | 0.708 (0.007) | 1.145 (0.010) | 1.293 (0.003) | 1.06631 |
| JOAO + TDL$_{atom}$ | 1.086 (0.013) | 0.713 (0.004) | 1.120 (0.007) | 1.265 (0.008) | 1.04613 |
| GraphCL | 1.127 (0.023) | 0.722 (0.004) | 1.143 (0.015) | 1.317 (0.010) | 1.07725 |
| GraphCL + TDL$_{atom}$ | 1.082 (0.008) | 0.707 (0.009) | 1.113 (0.007) | 1.294 (0.011) | 1.04943 |

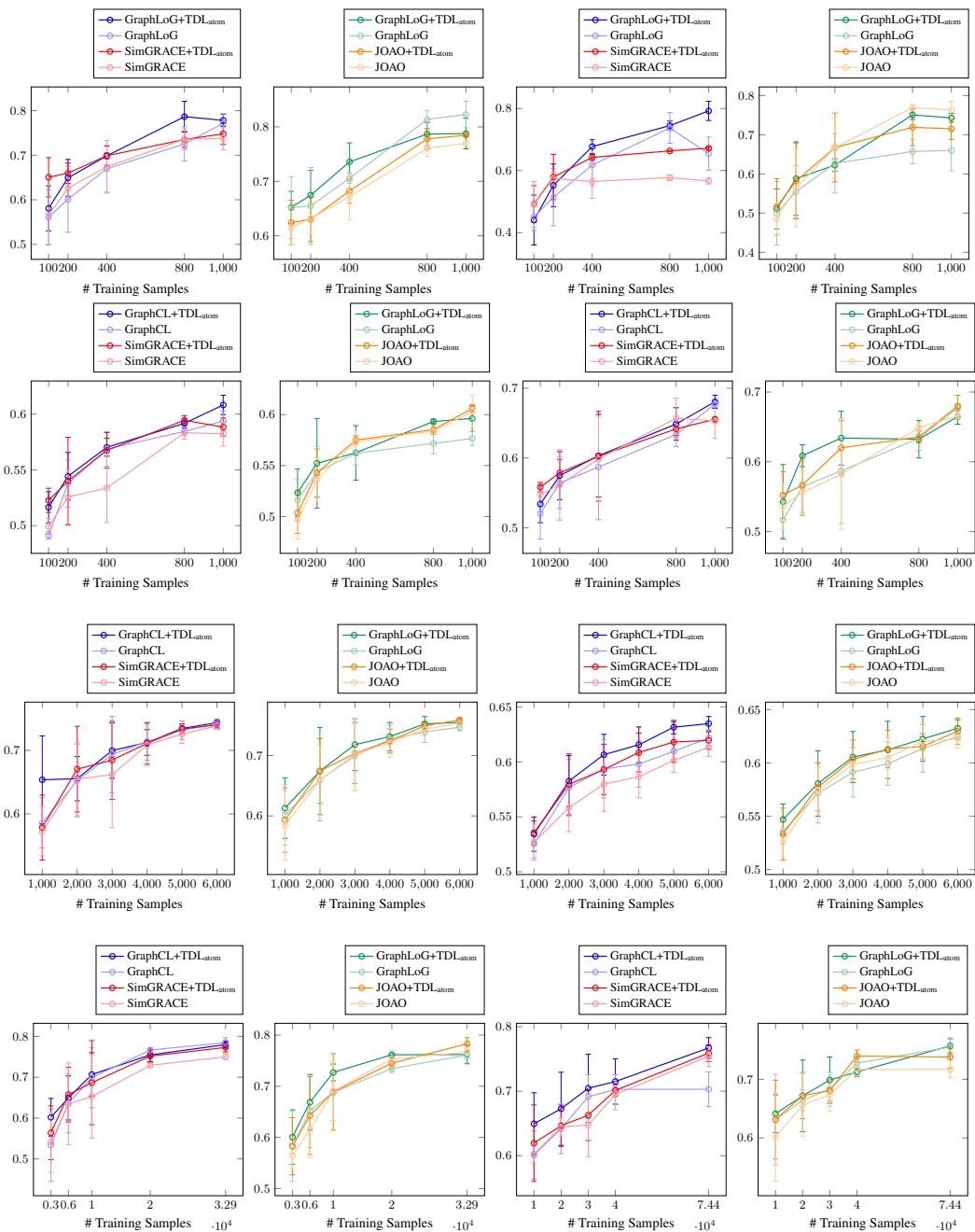

Figure 13: Performance over smaller datasets. Each datapoint is the mean ROC-AUC of 10 different splits while keeping scaffolds, with standard deviations shown as error bars. From the upper left to the lower right: Bace, ClinTox, Sider, BBBP, Tox21, ToxCast, HIV, MUV.

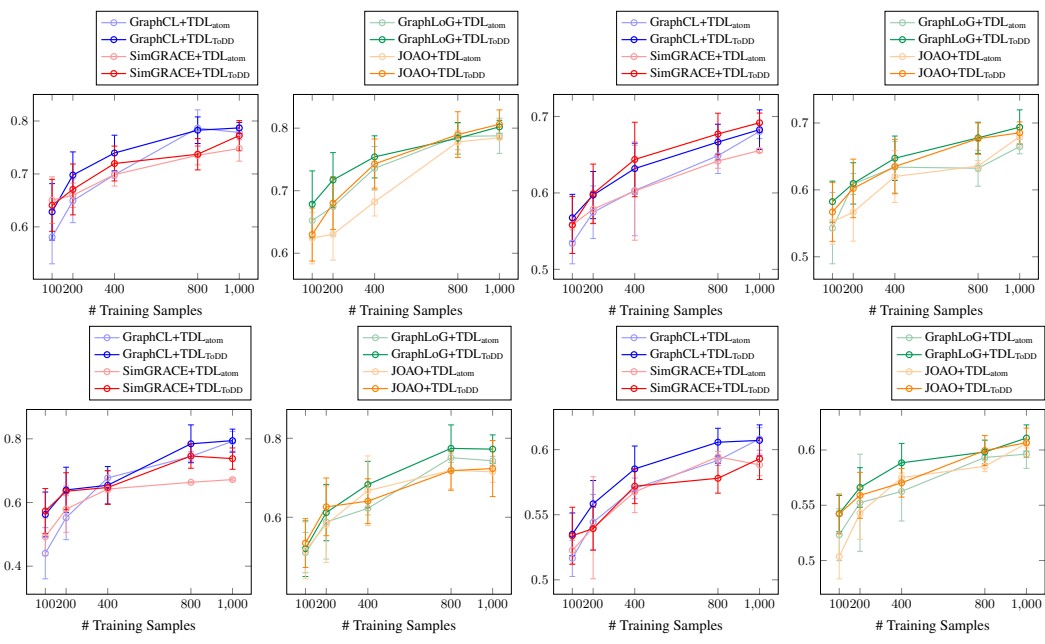

Figure 14: Performance over smaller datasets. Each datapoint is the mean ROC-AUC of 10 different splits while keeping scaffolds, with standard deviations shown as error bars. From the upper left to the lower right: Bace, BBBP, Clintox, Sider.

# E  Low-Data Scenario

Here we perform a more detailed experiment to illustrate how model performances change with increasing numbers of training samples. We fine-tune multiple models (GraphCL, JOAO, GraphLoG and SimGRACE) and TDL versions on 8 molecular property prediction datasets with training sets of different size. The experiment of each low-data size is run with 10 different subsets of the original training sets while keeping scaffolds. Figure 13 and Figure 14 show that TDL often yields remarkable improvements. Specifically, we observe that TDL's impact seems to depend on two factors: (1) in how far it is able to resolve issues of the baseline (e.g., the dimensional collapse of GraphCL), and (2) in how far the PIs used for TDL "suit" the dataset considered (see also Figure 8). We observe that the latter yields a task which needs to be solved together with domain experts (i.e., finding the best knowledge to use to create PIs).

# F   Unsupervised Learning

**Datasets.** In the unsupervised context, we adopted various molecular benchmarks from TUDataset [Morris et al., 2020], including NCI1, MUTAG, PROTEINS, and DD. Table 15 shows statistics for datasets.

Table 15: Summary for molecular datasets from the benchmark TUDataset.

| Dataset | # Molecules | # Class | Avg. # Nodes | Avg. # Edges |
|---------|-------------|---------|--------------|--------------|
| NCI1 | 4,110 | 2 | 29.87 | 32.30 |
| MUTAG | 188 | 2 | 17.93 | 19.79 |
| PROTEINS | 1,113 | 2 | 39.06 | 72.82 |
| DD | 1,178 | 2 | 284.32 | 715.66 |

**Configurations.** For the unsupervised graph classification task, we first train a representation model contrastively using unlabeled data, then fix the representation model and train the classifier using labeled data. Following GraphCL [You et al., 2020], we employ a 5-layer GIN with a hidden size of 128 as our representation model and utilize an SVM as our classifier. The GIN is trained with a batch size of 128 and a learning rate of 0.001. Regarding graph representation learning, models are trained for 20 epochs and tested every 10 epochs. We conduct a 10-fold cross-validation on every dataset. For each fold, we utilize 90% of the total data as the unlabeled data for contrastive pre-training and the remaining 10% as the labeled testing data. Every experiment is repeated 5 times using different random seeds, with mean and standard deviation of accuracies (%) reported. We apply TDL on GraphCL and SimGRACE. The temperature parameter $\tau$ for TDL here is chosen from $\{0.1, 0.2, 0.5, 1.0, 10.0\}$. Besides adding a TDL to the loss function in CL approaches, we did not make any modifications to the models.

**Results.** Table 16 displays the comparison among different models for unsupervised learning, revealing that a general performance improvement is achieved with the TDL. We also compute the normalized Euclidean distance of embeddings from these unsupervised datasets, and the histogram is illustrated in Figure 15. The findings are consistent with our prior observations in transfer learning; however, the impact may not be as substantial due to the smaller number of training samples.

Table 16: Results of unsupervised learning. For each datasets, we report the mean (and standard deviation) accuracies of 5 seeds.

|  | NCI1 | MUTAG | DD | PROTEINS |
|--|------|-------|-----|----------|
| GraphCL | 77.87 (0.41) | 86.80 (1.34) | 78.62 (0.40) | 74.39 (0.45) |
| GraphCL + TDL$_{atom}$ | **80.06** (0.37) | **89.12** (0.79) | **79.88** (0.47) | **75.59** (0.48) |
| SimGRACE | 79.12 (0.44) | 89.01 (1.31) | 77.44 (1.11) | 75.35 (0.09) |
| SimGRACE + TDL$_{atom}$ | **80.08** (0.31) | **89.47** (1.09) | **79.01** (0.84) | **75.39** (0.57) |

# G   Ablation Studies

**Different Filtrations - Geometry.** Firstly, we consider detailed ablations of the filtration functions used in the graph filtration step of TAE$_{ahd}$. That is, we use individual PIs (i.e., each PI is/was constructed based on a separate filtration function) instead of concatenation as reconstruction targets for TAE$_{ahd}$. Consequently, we obtain 3 individual TAEs based on different PIs. We also conduct an experiment that involves integrating these 3 TAEs during the fine-tuning of downstream tasks. More specifically, we individually fine-tune these models, concatenate their embeddings after the readout, and finally project through a projection head for downstream task prediction. As demonstrated in Table 17, concatenation during fine-tuning (FT) does indeed combine the advantages of individual filtrations. These filtrations are very generic and the method is flexible in terms of what is useful downstream. That is, this approach would allow domain experts to flexibly use their own filtrations in combinations and together with existing methods, in dependence on the specific downstream data. This also suits the well-known fact that specific downstream data may benefit from specific domain knowledge.

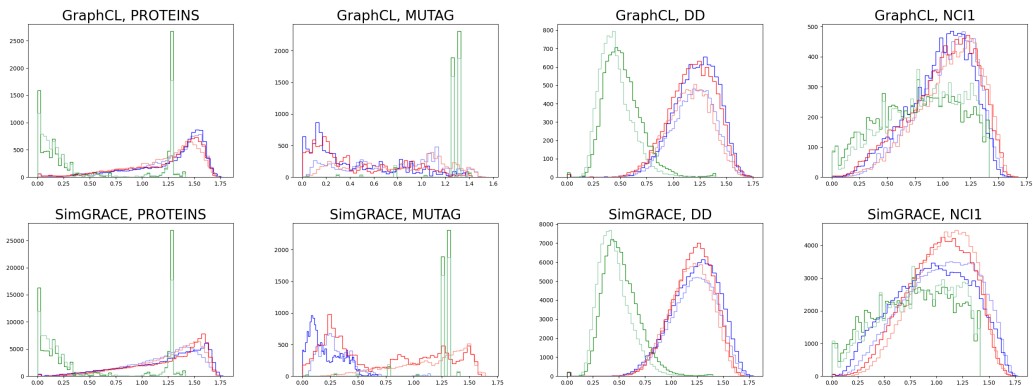

Figure 15: Evaluating alignment in unsupervised datasets. Dark/light shades denote positive/negative pairs, respectively. Blue illustrates the embedding of baselines, red depicts the embedding of baselines+TDL$_{atom}$, and green represents PI$_{atom}$.

For TDL, the filtration based on atom number provided overall best performance; we ran all experiments on PIs concatenated as above as well. We plan to investigate more in this direction in the future.

**Different Filtrations - Domain Knowledge.** Furthermore, we show results about incorporating different kinds of domain knowledge, here, atomic radius and electron affinity, as filtration functions. For TAE, we conduct two experiments: (1) We pre-train TAE using individual PIs generated by atomic radius and electron affinity as reconstruction targets; (2) We concatenate these 5 PIs (with the previous 3 PIs) as reconstruction targets. For TDL, we use these two individual PIs as topological fingerprints and apply TDL on GraphCL. Table 17 shows that these two domain knowledge-based filtration functions achieve commendable average performance. Yet, in summary, we do not observe considerable differences. Our hypothesis is that, similar to the atomic number, these kinds of domain knowledge capture the difference between the atoms, and hence are not too different overall. Based on this investigation, we plan to experiment with more complex sources of domain knowledge (e.g., the polarity of atoms, which is important information for the possible relationships between atoms).

**Different Filtrations - Multiple Dimensions**. Here we consider the multidimensional Vietoris-Rips (multi-VR) filtration [Demir et al., 2022]. And we use the atomic number, heat kernel signature with $t = 0.1$ and node degree as filtration functions in the first dimension, respectively. Then we compute PDs and then vectorize them to get 3 PIs. Finally, we concatenate these 3 PIs as reconstruction targets to pre-train TAE.

**No Filtrations.** Please observe first that our study's (initial) research question has been the exploration of the usefulness of PH and topological fingerprints for molecular SSL. Based on our investigation (see also Figure 8), there is one very interesting outcome of our research and follow-up research question related to TDL: TDL is an instance of a more general proposal "XDL", a novel loss function for pre-training which is exploits distances between training samples. We conducted some initial experiments only using the Tanimoto distance between standard ECFP fingerprints and obtain decent results; see Table 17. In future research, we will investigate the generality and potential impact of XDL more (e.g., combining ECFPs and topological fingerprints).

**Different Vectorizations of PDs**. Persistence Vectorizations transform the obtained Persistent Homology information (PDs) into either a functional or a feature vector form. This representation is more suitable for ML tools compared to PDs. In our study, we employ Persistence Images for vectorization. Apart from this, we also explored other stable vectorizations for TAE, such as Persistence Landscapes [Bubenik et al., 2015] and Silhouettes [Chazal et al., 2014]. As shown in Table 18, ablation experiments demonstrate that Persistence Images outperform the other two vectorization techniques.

**Hyperparameters Analysis**. We further perform a sensitivity analysis on the parameter $\lambda$ of GraphCL + TDL$_{ToDD}$ across 8 downstream molecular tasks. The results are presented in Table 19. It

is noteworthy that as $\lambda$ decreases, the weight of TDL also decreases, leading to a noticeable decline in performance. This observation underscores the effectiveness of TDL.

Table 17: Ablation study results for 8 molecular property prediction tasks on ZINC15. For each downstream task, we report the mean (and standard deviation) ROC-AUC of 10 seeds.

| | Tox21 | ToxCast | Sider | ClinTox | MUV | HIV | BBBP | Bace | Average |
|---|---|---|---|---|---|---|---|---|---|
| No pretrain | 74.6 (0.4) | 61.7 (0.5) | 58.2 (1.7) | 58.4 (6.4) | 70.7 (1.8) | 75.5 (0.8) | 65.7 (3.3) | 72.4 (3.8) | 67.15 |
| TAE$_{ahd}$ | 75.2 (0.8) | 63.1 (0.3) | 61.9 (0.8) | 80.6 (1.9) | 74.6 (1.8) | 73.5 (2.1) | 67.5 (1.1) | 82.5 (1.1) | 72.36 |
| TAE$_{atom}$ | 74.6 (0.3) | 61.8 (0.6) | 62.5 (0.7) | 71.0 (3.8) | 75.7 (2.1) | 75.5 (0.8) | 67.0 (1.1) | 78.6 (1.3) | 70.84 |
| TAE$_{hks}$ | 75.5 (0.3) | 62.7 (0.3) | 59.0 (0.6) | 79.3 (3.0) | 75.5 (1.4) | 73.5 (1.2) | 68.6 (0.5) | 82.5 (0.8) | 72.08 |
| TAE$_{degree}$ | 75.6 (0.6) | 62.8 (0.5) | 58.2 (0.8) | 80.0 (1.3) | 70.7 (2.0) | 73.4 (1.6) | 68.1 (0.8) | 81.9 (0.6) | 71.34 |
| TAE$_{Concatenation during FT}$ | 75.7 (0.6) | 64.2 (0.4) | 61.9 (1.2) | 80.2 (2.1) | 74.4 (2.9) | 75.1 (1.1) | 68.1 (0.8) | 82.1 (1.1) | 72.71 |
| TAE$_{Atomic radius}$ | 76.5 (0.6) | 63.3 (0.7) | 63.2 (1.4) | 76.2 (2.6) | 71.9 (2.7) | 74.3 (1.3) | 67.4 (1.3) | 80.9 (1.7) | 71.71 |
| TAE$_{Electron affinity}$ | 75.6 (0.6) | 63.0 (0.7) | 64.0 (0.7) | 78.5 (1.9) | 72.6 (2.8) | 76.3 (1.6) | 68.9 (1.0) | 81.5 (1.3) | 72.55 |
| TAE$_{Concatenation of 5 PIs}$ | 75.9 (0.4) | 63.8 (0.3) | 62.5 (0.6) | 81.6 (2.4) | 75.6 (1.4) | 74.2 (0.9) | 66.6 (0.9) | 82.7 (1.2) | 72.86 |
| TAE$_{Multi-VR filtration}$ | 75.8 (0.6) | 63.4 (0.6) | 62.0 (0.7) | 76.8 (2.6) | 75.1 (1.2) | 76.4 (0.6) | 69.9 (0.9) | 81.7 (1.3) | 72.64 |
| TAE$_{ToDD}$ | 76.8 (0.9) | 64.0 (0.5) | 61.9 (0.8) | 79.3 (3.6) | 75.8 (3.2) | 75.9 (1.1) | 70.4 (0.8) | 81.6 (1.4) | 73.22 |
| GraphCL | 75.0 (0.3) | 62.8 (0.2) | 60.1 (1.3) | 78.9 (4.2) | 77.1 (1.0) | 75.0 (0.4) | 67.5 (3.3) | 68.7 (7.8) | 70.64 |
| GraphCL + TDL$_{atom}$ | 75.3 (0.4) | 64.4 (0.3) | 61.2 (0.6) | 83.7 (2.7) | 75.7 (0.8) | 78.0 (0.9) | 70.9 (0.6) | 80.5 (0.8) | 73.71 |
| GraphCL + TDL$_{Atomic radius}$ | 74.3 (0.3) | 63.2 (0.3) | 61.3 (0.4) | 85.5 (1.5) | 77.6 (2.4) | 77.3 (1.3) | 70.7 (0.6) | 74.3 (1.3) | 73.02 |
| GraphCL + TDL$_{Electron affinity}$ | 75.1 (0.3) | 63.3 (0.4) | 60.2 (0.6) | 84.9 (3.0) | 76.0 (1.4) | 78.8 (1.9) | 70.1 (0.6) | 79.5 (1.1) | 73.50 |
| GraphCL + TDL$_{ToDD}$ | 75.2 (0.7) | 64.2 (0.3) | 61.5 (0.4) | 85.2 (1.8) | 75.9 (2.1) | 77.9 (0.8) | 69.9 (0.9) | 81.2 (1.9) | 73.88 |
| GraphCL + TDL$_{ECFP}$ | 75.3 (0.4) | 64.0 (0.2) | 61.1 (0.6) | 82.6 (2.8) | 75.0 (1.1) | 76.4 (0.9) | 70.1 (0.7) | 78.3 (1.7) | 72.85 |

Table 18: Comparison of the different vectorizations.

| | Tox21 | ToxCast | Sider | ClinTox | MUV | HIV | BBBP | Bace | Average |
|---|---|---|---|---|---|---|---|---|---|
| TAE$_{ahd}$ (Persistence Images) | 75.2 (0.8) | 63.1 (0.3) | 61.9 (0.8) | 80.6 (1.9) | 74.6 (1.8) | 73.5 (2.1) | 67.5 (1.1) | 82.5 (1.1) | 72.36 |
| TAE$_{ahd}$ (Persistence Landscapes) | 75.6 (0.5) | 63.9 (0.5) | 59.5 (0.8) | 73.5 (2.9) | 73.6 (0.9) | 74.0 (1.4) | 69.7 (1.6) | 79.8 (1.4) | 71.21 |
| TAE$_{ahd}$ (Silhouettes) | 74.7 (0.6) | 63.4 (0.6) | 58.4 (0.7) | 74.1 (2.4) | 72.2 (1.6) | 75.8 (1.4) | 68.9 (0.9) | 79.5 (1.4) | 70.88 |

Table 19: Sensitivity analysis on the parameter $\lambda$.

| GraphCL + TDL$_{ToDD}$ | Tox21 | ToxCast | Sider | ClinTox | MUV | HIV | BBBP | Bace | Average |
|---|---|---|---|---|---|---|---|---|---|
| $\lambda = 0.1$ | 74.2 (0.8) | 63.1 (0.2) | 60.8 (0.6) | 76.4 (5.8) | 73.3 (1.7) | 77.2 (1.6) | 70.2 (0.2) | 71.6 (0.9) | 70.85 |
| $\lambda = 0.5$ | 74.0 (0.5) | 63.2 (0.3) | 60.0 (0.4) | 78.6 (0.1) | 73.6 (1.4) | 77.2 (0.6) | 69.9 (0.3) | 76.2 (1.7) | 71.59 |
| $\lambda = 1.0$ | 75.2 (0.7) | 64.2 (0.3) | 61.5 (0.4) | 85.2 (1.8) | 75.9 (2.1) | 77.9 (0.8) | 69.9 (0.9) | 81.2 (1.9) | 73.88 |
| $\lambda = 2.0$ | 74.8 (0.3) | 63.6 (0.3) | 61.7 (0.7) | 85.8 (1.7) | 74.9 (0.8) | 77.2 (1.0) | 71.0 (0.7) | 81.8 (1.2) | 73.85 |

**Different GNN Backbones**. We apply TDL on GraphCL. As shown in Table 20, we *verify the generality of TDL* by experimenting with three popular GNN models: GIN [Xu et al.], GCN [Kipf and Welling], and GraphSAGE [Hamilton et al., 2017]. Pre-training with GIN achieves the best performance across datasets, hence our results confirm what has been observed in related works.

Table 20: Comparison of the pre-training gains, represented in terms of averaged ROC-AUC (%), obtained by using different GNN architectures on 8 datasets.

| Model | GCN | GIN | GraphSAGE |
|---|---|---|---|
| No pretrain | 68.77 | 67.15 | 68.32 |
| GraphCL | 69.32 | 70.78 | 70.08 |
| GraphCL + TDL$_{atom}$ | 72.53 | 73.71 | 71.16 |

# H PIs w/o SSL

Here we provide initial results for a comparison with 2 non pre-training ML models, i.e. support vector machine (SVM) and XGB [Chen and Guestrin, 2016] to test the regular prediction on 5 molecular property prediction tasks using PIs as topological fingerprints. And we also conduct smaller data experiments on SVM compared with TAE and TAE+ContextPred over 4 downstream datasets. Note that we performed a hyperparameter search to produce these ML results. As can be observed in Table 21 and Figure 16, the SSL methods provide large benefits. We are certainly aware of the fact that there are more complex methods to use these PIs, yet *this basic comparison gives a good hint of how SSL based on PH compares to supervised methods more generally.*

Table 21: Results of non pre-training method for molecular property prediction tasks on ZINC15, ROC-AUC (%).

| | Tox21 | Sider | ClinTox | BBBP | Bace |
|---|---|---|---|---|---|
| SVM | 53.9 | 53.8 | 60.1 | 56.6 | 72.3 |
| XGB | 64.7 | 58.5 | 55.7 | 61.6 | 73.8 |
| ContextPred | 75.7 | 60.9 | 65.9 | 68.0 | 79.6 |
| TAE$_{ahd}$ + ContextPred | **76.4** | **62.0** | **74.6** | **68.9** | **80.7** |

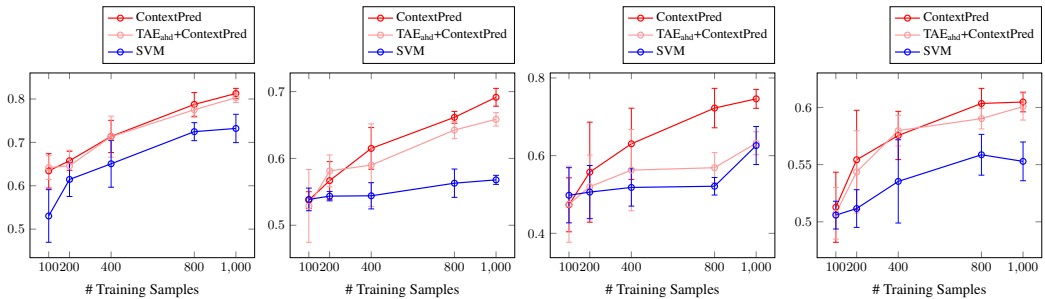

Figure 16: Performance of SVM over smaller datas. Each datapoint is the mean ROC-AUC of 10 different splits while keeping scaffolds, with standard deviations shown as error bars. From the left to the right: Bace, BBBP, ClinTox, Sider.

# I Theoretical Analysis of TDL

In this section, we delve into a theoretical analysis of TDL, which was initially introduced in Section 3.2. We encourage readers to refer to Section 3.2 for any specific notation, terminology, or formulation details.

Recall TDL is defined as follows:

$$\mathcal{L}_{\text{TDL}_n} = \frac{1}{N-1} \sum_{\substack{m \in [1,N], \\ m \neq n}} -\log \frac{\exp\left(sim\left(z_n, z_m\right)/\tau\right)}{\sum_{\substack{k \in [1,N], \\ k \neq n}} \mathbb{I}_{[dis(I_n, I_k) \geq dis(I_n, I_m)]} \cdot \exp\left(sim\left(z_n, z_k\right)/\tau\right)}$$

For the sake of convenience, we re-index the subscripts of samples, ensuring that the subscripts $d_i$ satisfy the sorted distances from the n-th sample, i.e., $dis(I_n, I_n) \leq dis(I_n, I_{d_1}) \leq dis(I_n, I_{d2}) \leq dis(I_n, I_{d3}) \leq ... \leq dis(I_n, I_{d_{N-1}})$. Let $sim(z_n, z_m) = z_n z_m$ to streamline the derivations. We thus obtain

$$\mathcal{L}_{\text{TDL}_n} = \frac{1}{N-1} \sum_{j \in [1, N-1]} - \log \frac{e^{(z_n \cdot z_{d_j}/\tau)}}{\sum_{k \in [j, N-1]} e^{(z_n \cdot z_{d_k}/\tau)}}$$

We then analyze the gradients with respect to different samples.

**Lemma 1.** *The gradient of $TDL_n$ with respect to the sample $d_i$ in latent space is formulated as:*

$$\nabla_{z_{d_i}} \mathcal{L}_{\text{TDL}_n} = \frac{1}{N-1} \left( \sum_{j \in [1, i]} \frac{e^{z_n \cdot z_{d_i}/\tau}}{\sum_{k \in [j, N-1]} e^{z_n \cdot z_{d_k}/\tau}} - 1 \right) \cdot \frac{z_n}{\tau}.$$

*In particular, $\nabla_{z_{d_1}} \mathcal{L}_{\text{TDL}_n} \propto -z_n$ and $\nabla_{z_{d_{N-1}}} \mathcal{L}_{\text{TDL}_n} \propto z_n$.*

*Proof.* The results follow from direct computation. Let $\mathcal{L}_{\text{TDL}_{n, d_j}} = -\log \frac{e^{(z_n \cdot z_{d_j}/\tau)}}{\sum_{k \in [j, N-1]} e^{(z_n \cdot z_{d_k}/\tau)}}$, then we have:

$$\mathcal{L}_{\text{TDL}_n} = \frac{1}{N-1} \sum_{j \in [1, N-1]} - \log \frac{e^{(z_n \cdot z_{d_j}/\tau)}}{\sum_{k \in [j, N-1]} e^{(z_n \cdot z_{d_k}/\tau)}} = \frac{1}{N-1} \sum_{j \in [1, N-1]} \mathcal{L}_{\text{TDL}_{n, d_j}}$$

Next, we discuss $\nabla_{z_{d_i}} \mathcal{L}_{\text{TDL}_{n, d_j}}$:

$$\nabla_{z_{d_i}} \mathcal{L}_{\text{TDL}_{n, d_j}} = \begin{cases} 0 & i < j \\ \left( \frac{e^{z_n \cdot z_{d_i}/\tau}}{\sum_{k \in [j, N-1]} e^{z_n \cdot z_{d_k}/\tau}} - 1 \right) \cdot \frac{z_n}{\tau} & i = j \\ \left( \frac{e^{z_n \cdot z_{d_i}/\tau}}{\sum_{k \in [j, N-1]} e^{z_n \cdot z_{d_k}/\tau}} \right) \cdot \frac{z_n}{\tau} & i > j \end{cases}$$

Therefore, we conclude:

$$\nabla_{z_{d_i}} \mathcal{L}_{\text{TDL}_n} = \frac{1}{N-1} \nabla_{z_{d_i}} \sum_{j \in [1, N-1]} \mathcal{L}_{\text{TDL}_{n, d_j}} = \frac{1}{N-1} \left( \sum_{j \in [1, i]} \frac{e^{z_n \cdot z_{d_i}/\tau}}{\sum_{k \in [j, N-1]} e^{z_n \cdot z_{d_k}/\tau}} - 1 \right) \cdot \frac{z_n}{\tau}$$

Then we have:

$$\nabla_{z_{d_1}} \mathcal{L}_{\text{TDL}_n} = \frac{1}{N-1} \left( \frac{e^{z_n \cdot z_{d_1}/\tau}}{\sum_{k \in [1, N-1]} e^{z_n \cdot z_{d_k}/\tau}} - 1 \right) \cdot \frac{z_n}{\tau}$$

The quantity $\left( \frac{e^{z_n \cdot z_{d_1}/\tau}}{\sum_{k \in [1, N-1]} e^{z_n \cdot z_{d_k}/\tau}} - 1 \right) < 0$ is a strictly negative scalar, allowing us to conclude the derivative $\nabla_{z_{d_1}} \mathcal{L}_{\text{TDL}_n}$ is proportional to $-z_n$.

Similarly, we have:

$$\nabla_{z_{d_{N-1}}} \mathcal{L}_{\text{TDL}_n} = \frac{1}{N-1} \left( \sum_{j \in [1, N-2]} \frac{e^{z_n \cdot z_{d_{N-1}}/\tau}}{\sum_{k \in [j, N-1]} e^{z_n \cdot z_{d_k}/\tau}} \right) \cdot \frac{z_n}{\tau}$$

Since $\left( \sum_{j \in [1, N-2]} \frac{e^{z_n \cdot z_{d_{N-1}}/\tau}}{\sum_{k \in [j, N-1]} e^{z_n \cdot z_{d_k}/\tau}} \right) > 0$ we conclude in this case that the derivative $\nabla_{z_{d_{N-1}}} \mathcal{L}_{\text{TDL}_n}$ points in the direction $z_n$. □

The stability provided by certain topological fingerprints serves as a mathematically grounded, well-studied, and efficient proxy for stability with respect to graphs. TDL as our objective is based on the stability of these topological fingerprints. Consequently, for each sample, we can determine a ranking of all other samples. Lemma 1 provides the gradient with respect to different samples. Specifically, samples that are distant from sample $n$ in the topological space point in the direction $-z_n$ (negative gradient direction), while those close to sample $n$ point in the direction $z_n$. Consequently, molecules with more similar topological features are brought closer to each other in the embedding space.

## J    Potential of PH for Molecular ML

Persistent homology (PH) has emerged as a crucial method in topological data analysis, demonstrating significant potential in decoding complex patterns within molecular data, including protein prediction [Cang and Wei, 2017a,b, Nguyen et al., 2019, Xia and Wei, 2014, Liu et al., 2022b], virtual screening [Cang et al., 2018], and drug design [Liu et al., 2021]. The many works on PH in chemistry show that the domain is convinced by the usefulness of topological fingerprints already, and we therefore believe that this kind of knowledge should be investigated in the context of SSL, assuming that the latter becomes more important with the advancement of foundation models.

## K    Limitations & Broader Impact

**Broader Impact.** Graph data has a pervasive impact in many diverse fields and the popularity certainly increases the danger of an application of the models with negative effects (even if just due to ignorance). As pointed out in this work w.r.t. smaller data, we have to be aware of the assumptions we make in our research to inform readers on how the results translate to the real world. Our goal is to advance biomedicine, chemistry, and material science, where our research hopefully provides benefits one day, for both the current society and future generations.

**Limitations.** We have tried our best in addressing common issues, by providing a very detailed analysis, by considering TDL on top of various baselines, by also reporting mixed results, and by providing all code. However, due to time and resource constraints, and the broadness of our study, we were not able to experiment with a variety of topological fingerprints, analyze individual baselines (e.g., the dimensional collapse) in more detail, or complete the theoretical investigations.

