# OpenReview forum: "Improving Self-supervised Molecular Representation Learning using Persistent Homology"
_NeurIPS.cc/2023/Conference — NeurIPS 2023 poster_

### Official Review · Reviewer_recC · 2023-06-22

**Soundness:** 3 good
**Presentation:** 3 good
**Contribution:** 3 good
**Rating:** 7
**Confidence:** 4

**Summary:**

In this manuscript, the authors have developed an interesting self-supervised learning model, by the incorporation of persistent homology into contrastive learning module. More specifically, a special topological distance based contrastive loss is proposed. The model is novel, and the results are very promising. However, I have some concerns about the persistent homology analysis part.


**Strengths:**

The authors have developed an interesting self-supervised learning model, by the incorporation of persistent homology into contrastive learning module. More specifically, a special topological distance based contrastive loss is proposed. The model is novel, and the results are very promising.

**Weaknesses:**

The PH model is not explained clearly.

**Questions:**

1)	As a powerful tool in topological data analysis, persistent homology (PH) has demonstrated great power in molecular data analysis, PH-based molecular descriptors and fingerprints have already been extensively tested on various benchmark datesets and shown better performance than not only traditional molecular descriptors, such as ECFP, Morgan, daylight, etc, but also many deep learning models. Many related important references are not discussed in the paper, such as

Z. X. Cang, Lin Mu and Guo-Wei Wei, Representability of algebraic topology for biomolecules in machine learning based scoring and virtual screening, PLOS Computational Biology, 14(1), e100592 (2018).

Z. X. Cang and Guo-Wei Wei, TopologyNet: Topology based deep convolutional and multi-task neural networks for biomolecular property predictions, PLOS Computational Biology, 13(7), e1005690 (2017).

Z. X. Cang and Guo-Wei Wei, Element specific persistent homology for the analysis and prediction of protein folding stability upon mutation, Bioinformatics,  33, 3549-3557 (2017).

Z. X. Cang, Lin Mu and Guo-Wei Wei, Representability of algebraic topology for biomolecules in machine learning based scoring and virtual screening, PLOS Computational Biology,  14(1), e100592 (2018).

Duc Duy Nguyen, Zixuan Cang, Kedi Wu, Menglun Wang, Yin Cao and Guo-Wei Wei, Mathematical deep learning for pose and binding affinity prediction and ranking in D3R Grand Challenges, Journal of Computer Aided Molecular Design, 33, 71-82  (2019).

Duc Nguyen, Zixuan Cang, and Guo-Wei Wei, A review of mathematical representations of biomolecular data, Physical Chemistry Chemical Physics, 22, 4343-4367 (2020).

Xiang Liu, Xiangjun Wang, Jie Wu, and Kelin Xia, "Hypergraph based persistent cohomology (HPC) for molecular representations in drug design." Briefings In Bioinformatics, 22 (5), bbaa411 (2021)

Xiang Liu, Huitao Feng, Jie Wu, and Kelin Xia, "Dowker complex based machine learning (DCML) models for protein-ligand binding affinity prediction." PLOS Computational Biology, 18(4), e1009943 (2022)

Chi Seng Pun, Si Xian Lee, and Kelin Xia, "Persistent-homology-based machine learning: a survey and a comparative study." Artificial Intelligence Review, (2022)

D. Vijay Anand, Qiang Xu, Junjie Wee, Kelin Xia, and Tze Chien Sum, "Topological feature engineering for machine learning based halide perovskite materials design", npj Computational Materials, 8 (203) (2022)

2) Mathematically, the filtration process from PH will only generate a nested sequence of simplicial complexes! In Figure 1, the plotted “chemical subgraphs” (during the filtration process) are not the general simplicial complexes (Vietoris-Rips complex or Alpha complex), as there are double bonds. Note that a double bond is illustrated as two edges which appear simultaneous among two vertices. This representation is not mathematically rigorous! The authors can use the common graph but add edge features to denote the double bond! Note that it is possible to use cellular complex to represent “double bonds”, but their persistent homology will be different!

3) Page 3, line 100, their filtration ends with the original graph. In this way, their Betti_1 bars will never die, as there are NO 2-simplexes in their model? The authors are suggested to double check this setting. More discussion will be given below.

4) Page 3, line 106, the authors mention that they use $H_k (G_i)$? They should specify what is the range for the integer $k$. Further, in their topological loss, do they consider both the Betti_0 and Betti_1, or just Betti_0/Betti_1?

5) Page 5, line 209-212, “The fingerprints are usually constructed in a way so that they capture information about the molecular graph structure and sometimes additional domain knowledge; even if they do not capture the entire complexity of the molecules, they represent some, probably important aspects.” Even though PH models are important, they can only characterize homological information, such as individual components, circles, voids, cavities. In biomolecular, the Betti_0 is usually related to covalent bonds, Betti_1 is related to pentagon (sugar ring) and hexagon (benzene ring). These findings have already been widely known, many references can be found,

Kelin Xia, Xin Feng, Yiying Tong and Guo-Wei Wei, "Persistent homology for the quantitative prediction of fullerene stability." Journal of Computational Chemistry, 36, 408-422(2015).

Kelin Xia and Guo-Wei Wei, "Persistent homology analysis of protein structure, flexibility and folding." International Journal for Numerical Methods in Biomedical Engineering, 30(8), 814-844(2014)

Zhenyu Meng, D Vijay Anand, Yunpeng Lu, Jie Wu, and Kelin Xia, "Weighted persistent homology for biomolecular data analysis." Scientific Report, 10 (1), 1-15 (2020)

The authors are suggested to add an example or some more discussions to explain what the information is captured in their PH models. Further, the general filtration is usually based on atomic distance or some special weighted distances, in this way, higher-dimensional simplicial complexes are generated!  And both Betti_0, Betti_1, and Betti_2 have their clear biological meanings! If the filtration process ends with the original graph (as illustrated in Page 3, line 100), there are no Betti_2 information, and Betti_1 bars will never die! The authors are suggested to compare with these existing approaches to show their advantages or add some more discussions about the difference.


**Limitations:**

Missing many important related literature. The advantage of their filtration process is not clear.

---

> ### Author Rebuttal · Authors · 2023-08-09
>
> **Thank you for carefully checking our theory part!**
>
> We are sorry for the unnecessary confusion caused by the missing details.
> We hope that the additional explanations resolve the points mentioned, esp. also Q1 and Q5, which are part of the global reply. In particular, please note that we did not intend to propose a particular filtration function
> rather give an example. We consider this to better be left to the expertise of PH experts and dependent on the actual application.
>
> **Clarification on Theoretical Background**
>
> - Q2: The “double bonds” in Figure 1 serve as a graphical illustration of the edge features rather than part of the actual filtration. In our experiments, the presence of a double bond is also treated as an edge feature, which is clearly stated in Appendix A. We realize that this is confusing in the context where the filtration is introduced.
> - Q3: We intended to introduce a simplified scenario and give the more complex details about the version we actually used (an extended persistence module [1]) in the appendix, but later forgot to add a comment and the latter details. The extended persistence module [1] captures Betti_1 features, ensuring that they will also be killed in the end.
> - Q4: k is either 0 or 1. In TDL we use both in concatenation.
>
> We updated all these accordingly now.
>
> **Q5 Topological Features in Embeddings**
>
> In regular molecule representation learning, it is indeed possible to draw rather direct conclusions about what is or can be captured in the embeddings if we use very straightforward filtration methods (e.g., no multi-parameter filtrations, which have been shown to be very powerful recently). However, only the TAE baseline we consider actually is optimized for learning PH-based embeddings. TDL has a more abstract goal and uses the information captured by the topological fingerprints only indirectly, for regularization. We do not expect the learnt embeddings to explicitly capture topological features and therefore did not include such examples into the paper, since this might be misleading. We will add more discussion about the topic more generally (e.g., for TAE, and also topological views for CL could capture such features) to underline the potential PH may offer in SSL. This also fits well with the proposal of Reviewer XtrW to mention alternative, potentially useful architectures.
>
> ---------------------------------------------------------
> [1] Cohen-Steiner et al. "Extending persistence using Poincaré and Lefschetz duality." Foundations of Computational Mathematics 9.1 (2009): 79-103.

---

> > ### Comment · Reviewer_recC · 2023-08-15
> >
> > Thanks for the reply. I have no further comments.

---

> ### Author Response · Authors · 2023-08-15
> **Thank you so much!**
>
> This final confirmation is very helpful and highly appreciated.

---

### Official Review · Reviewer_XtrW · 2023-06-23

**Soundness:** 3 good
**Presentation:** 4 excellent
**Contribution:** 3 good
**Rating:** 6
**Confidence:** 4

**Summary:**

Paper uses self-supervised learning tools for graph representation learning by facilitating topological data analysis (TDA) methods. In particular, for molecular representation learning, the authors use persistent homology outputs to improve the embeddings obtained by GNNs. They evaluated their model in molecular property prediction problem, and consistently got performance improvements.

**Strengths:**

GNNs and TDA are both very successful and completely different methods in graph representation learning. In the past years, there are several approaches to integrate these two methods effectively. With this aim, the paper proposes a new way to use TDA output to improve node embeddings in GNNs by using contrastive learning ideas. The idea is novel and has a lot of room for improvement.

Molecular Representation Learning is a significant application area for graph representation learning. The authors applied their model in this domain, in particular, molecular property prediction. They obtain strong results on this important question.

The paper's experimental part and ML details are strong. The authors made an in-depth analysis of the model from various angles.

**Weaknesses:**

The experimental results (Table 4) do not show significant improvements in several cases.

The results only report the performance of internal models. It would be nice to see the comparison with the SOTA results on these datasets.

PH construction seems weak as it does not use clique complexes, and only uses nodes and edges in the filtration, i.e., the top dimension is set to be 1. This filtration are not commonly used in graphs as it reduces PH to only node and edge counting by using a simple Euler Characteristics argument. However, fortunately, this does not affect their performance in this setting since molecular graphs are planar and do not have loops of length 3, as all loops have length $\geq 5$. The authors should add a note for nonexperts that for molecular graphs, this trivial filtration setting is equivalent to the traditional clique complex setting for sublevel filtration because of the special structures of molecular graphs (no cycle of length 3). For TDL, PI is a good choice, but for TAE, it looks weird. To be used in such a loss function, there are better stable PH vectorizations, e.g., Silhouette, landscape.

**Questions:**

Instead of using PH output  in the loss function to improve GNN embeddings, did you consider directly combining them, e.g., by simply concatenating PIs with GNN embeddings? I know this is a completely different approach, but this is a more direct method to combine both outputs.

**Limitations:**

There are various choices to be made in several places for the model. On one side, it gives flexibility to adapt the model to different settings, but on the other hand, it needs expertise in several domains for fine-tuning.

---

> ### Author Rebuttal · Authors · 2023-08-09
>
> **Thank you for the insightful comments!**
>
> W1 and W2 are addressed in the global reply. We hope that especially our explanation there and the additional results clarify our proposal. The suggestion of comparing to SOTA also in our extra experiments (W2) was very helpful and underlines our contribution.
> Please let us know in case any questions are left!
>
>
> **W3 Clarification on PH Construction**
>
> This is a good point, and we added that to the paper. We admittedly dropped many details since we were afraid that too much theory on persistent homology may prevent readers and ML researchers from seeing and getting convinced of its most important features. In fact, to the best of our knowledge, the theory is only more recently considered in the field; e.g., in the context of graph representation learning, graph homology has been studied in more detail in [1].
>
> We did not intend to propose TAE as an architecture but rather as a means to compare to and obtain and estimate for the usefulness of PIs in an SSL setting, since we apply them for TDL (see also the response to Reviewer 9rP5, W1). We will mention this more clearly.
>
>
> **Q1 What about Concatenating PIs with GNN embeddings?**
>
> This is a valid architecture proposal, and we also think that using PH for constructing views deserves further study.
>
> Since we consider the distance-based approach we propose with TDL to offer unique advantages for existing models (i.e., by incorporating regularization in terms of the relations between the input graphs, which the models do not explicitly consider), the possibility to improve a number of those by applying our loss on top, and its potential for follow-up research (see also the response to Reviewer bcsi), we chose to focus on this one in this very initial paper.
>
> **Limitation: Variety of Design Choices**
>
> While this can be considered as a limitation, we think that it rather shows the potential of the research direction we propose. Further research is definitely needed before the possible architectures can be reliably used in practice. However, the many works on PH in chemistry (see also the references suggested by Reviewer recC) show that the domain is convinced by the usefulness of topological fingerprints already and we therefore believe that this kind of knowledge should be considered in SSL, assuming that the latter becomes more important with the advancement of foundation models.
>
> We now added some text about this to the paper to clarify the intention of our work, to motivate further study, and to also avoid confusion as it appeared in other reviews (e.g., that we recommend to use a specific filtration function, what we don't).
>
> ------------------------------------------------------------
> [1] Rieck, Bastian. "On the Expressivity of Persistent Homology in Graph Learning." arXiv preprint arXiv:2302.09826 (2023).

---

> > ### Comment · Reviewer_XtrW · 2023-08-13
> >
> > Thank you very much for your detailed answers and additional experiments. I have no further questions. Good luck with your submission.

---

> ### Author Response · Authors · 2023-08-14
> **Thank you for getting back to us!**
>
> We highly appreciate the careful and positive evaluation and also the encouraging response!

---

### Official Review · Reviewer_U3kd · 2023-06-28

**Soundness:** 3 good
**Presentation:** 2 fair
**Contribution:** 1 poor
**Rating:** 3
**Confidence:** 4

**Summary:**

This paper proposes two molecular self-supervised learning methods, which consists of fingerprint autoencoder and topological distance contrastive learning. The insight behind this paper is to utilize topological fingerprint as a supervision in self-supervised learning. Thus, the authors reconstruct the topological fingerprint of a given molecule with autoencoder and filter out similar molecules in negative views in contrastive learning based on the similarity in topological distance space. The experimental results show that their method improves previous baselines in various downstream tasks.

**Strengths:**

- The paper is well written and easy to understand.

- The experimental results are comprehensive; the authors considers several setups such as linear probing and fine-tuning.

**Weaknesses:**

- Lack of Novelty: Excluding similar molecules from negative sample set is already considered in [1]. Conceptually, the difference of TDL and [1] is that TDL utilizes PH and [1] utilizes ECFP fingerprint (I know that the loss of [1] is based on augmented molecules, but I think this does not make big difference). This limits the novelty of this paper.

- Table 1 does not support the effectiveness of proposed method: Correlating the distance in embedding space with the distance in corresponding PIs are not the main purpose of molecular representation learning. If PIs are indeed very important, then why should we use learned representation of proposed method? Can't we just utilize PIs as the molecular representation? In other words, Table 1 and Table 17 seem to contradict.

- Insufficient rationalization of the usage of PIs: In molecular domain, ECFP fingerprint is a widely applied molecular representation since it reflect the substructure-wise molecular information. Why should we use PIs in molecular representation learning?

- Flexibility of TDL: The authors insisted that TDL can be flexibly and efficient applied with any graph contrastive learning framework. However, any other two existing methods can be composed with each other to improve the performance. For example, ContextPred + GraphCL is possible and the flexibility is not the unique feature of TDL.

- Table 4 seems weak: TDL (or TAE) combined with existing method does improve the overall performance. However, Mole-BERT and SEGA shows better performance than the proposed method.

----Sorry for confusion. I added the reference.

[1] Improving Molecular Contrastive Learning via Faulty Negative Mitigation and Decomposed Fragment Contrast, Wang et al., JCIM 2022

**Questions:**

- In Table 4, why some baseline methods are combined with TAE while others are combined with TDL? Can't TAE and TDL combined jointly?

- How is the performance of TAE (or TDL) jointly trained with SEGA? Does this improve SEGA itself or SEGA + other method (e.g., SEGA + ContextPred)?

- I would be convinced with the experimental results if the authors compare "TAE + TDL" vs other methods (Please refer "Flexibility of TDL" in Weakness).

**Limitations:**

Yes. The authors addressed the limitations.

---

> ### Author Rebuttal · Authors · 2023-08-09
>
> **Thank you for pointing out this indeed very related paper!** We should not have missed such closely related work, and we hope that the below delimitation resolves some of the related issues pointed out in your review. In fact, the detailed comparison highlights the novelty of our work.
>
> W3 and Q2 are addressed in the global reply.
>
>
> **W1  Clarification on Novelty: Comparison to [1]**
>
> - **Our paper's focus.** We study the benefits of PH for molecular SSL and propose technology that is suitable for incorporating PH into SSL, rather than just a loss that excludes "false" negative examples.
> - **We do not consider views.** The fact that we focus on input examples represents a considerable difference. While the similarity between rather similar molecules, such as views, might be captured by ECFP, the latter turned out to be not as effective as PIs in TDL (Tab. 15). We hypothesize that this is due to the fact that input examples that are overall different but have structural similarities might have too similar ECFPs for a fine-grained distance regularization.
> - **We filter the graphs based on the distances.** We experimented with weighting as used in [1] in the beginning but our method showed better performance in our setting. The weighting might work with ECFP since these fingerprints are discrete and hence rather coarse grained, while it may give too confusing signals to the model with PIs.
> - **Pretraining Data.** [1] used ~10M molecules. We reran their model for a fair comparison.
> - **Results.** With TDL, we obtain similar improvement in fine-tuning and linear probing as with the other models we considered.
>
>     |           | Tox21      | ToxCast    | Sider      | MUV        | ClinTox    | HIV        | BBBP       | Bace       | Average |
>     | --------- | ---------- | ---------- | ---------- | ---------- | ---------- | ---------- | ---------- | ---------- | ------- |
>     | [1]       | 75.1 (0.7) | 63.5 (0.4) | 59.4 (1.0) | 74.7 (1.9) | 81.0 (2.6) | 77.3 (1.2) | 69.6 (1.2) | 77.3 (1.0) | 72.24   |
>     | [1] + TDL | 75.9 (0.6) | 63.7 (0.3) | 60.7 (0.8) | 75.1 (1.3) | 83.8 (1.9) | 76.7 (0.7) | 71.2 (0.9) | 78.5 (1.3) | 73.20   |
>     | [1]       | 68.8 (0.4) | 60.4 (0.3) | 57.5 (1.1) | 59.3 (2.2) | 73.3 (2.1) | 67.5 (0.7) | 63.8 (0.8) | 7.9 (1.2)  | 65.43   |
>     | [1] + TDL | 69.8 (0.3) | 61.1 (0.4) | 59.0 (0.4) | 61.7 (1.7) | 72.8 (1.0) | 69.6 (0.8) | 64.4 (0.4) | 74.7 (0.9) | 66.64   |
>
> **W2 & W5 On the Demonstration of Effectiveness**
>
> Assuming the filtration functions are carefully chosen and possibly include external knowledge, PIs will likely offer useful features for molecular representation learning (see also the references suggested by Reviewer recC). Nevertheless, our paper does not intend to compete with the existing body of graph SSL research and outperform specific SOTA approaches, but rather *improve* it by exploiting the unique features of PH in a complementary way. This is also why our initial work on the topic focuses on TDL rather than a custom, standalone CL-based loss, where PH could be used for constructing views, which would likely be a better model than the simple TAE. *Table 1 shows that TDL is effective in slightly moving the molecule embeddings in the embedding space towards the structure of the PI space*. Learning both embeddings that are rich in information, to fit multiple possible downstream scenarios, and a well-structured embedding space are especially important in SSL (which has more specific requirements than regular molecular representation learning).
>
> Re SOTA, please note our discussion of fine-tuning experiments in SSL in the global reply, our results with stronger filtrations, and the improvement we obtain for AD-GCL.
>
>
> **W4 "Flexibility" of TDL**
>
> It is true that other approaches can be combined as well. What we mean is that TDL covers a dimension which is not addressed by the regular models. Hence, "flexibly" is intended to convey the fact that it clearly adds a novel form of regularization which is likely effective.
> We did not want to cause confusion with that wording, which is indeed not crystal clear, we are definitely open to alternative suggestions.
>
>
> **Q1 & Q3 TAE + TDL?**
>
> Since TDL's objective is a direct consequence of TAE's objective we do not expect particular benefits of this combination. In the paper, we only ran TAE in combination with ContextPred to obtain an idea of its performance in such combinations, but this is more intended as a side experiment. In combination with other CL-based approach, TAE's objective would be rather strict in that it enforces direct embedding similarity, which might contradict the other model's objectives. TDL is more general (and in a certain sense flexible) in that it regularizes adaptively.
>
> As noted above (W2 & W5), standalone TDL is not designed for SOTA comparison since it is clearly missing CL representation power by not considering views. We hope that our explanations provide clarification in this regard and that our additional experimental results help to justify our contribution. In case there is doubt left, please let us know.
>
> For analysis purpose, we ran the experiments and the results largely match our expectations. Yet, standalone TDL is surprisingly effective.
> |                 | Tox21      | ToxCast    | Sider      | ClinTox    | MUV        | HIV        | BBBP       | Bace       | Average |
> | --------------- | ---------- | ---------- | ---------- | ---------- | ---------- | ---------- | ---------- | ---------- | ------- |
> | TDL  | 75.8 (0.5) | 62.1 (0.5) | 62.2 (0.9) | 79.1 (3.8) | 75.2 (2.3) | 76.9 (0.9) | 66.5 (1.8) | 78.4 (1.1) | 72.02   |
> | TAE+TDL         | 76.2 (0.3) | 62.9 (0.4) | 60.6 (1.0) | 81.7 (1.8) | 74.2 (1.5) | 76.2 (1.1) | 67.4 (0.8) | 83.0 (1.5) | 72.78   |
> | TAE+GraphCL+TDL | 76.0 (0.3) | 63.7 (0.4) | 62.6 (0.6) | 82.8 (2.3) | 75.4 (2.3) | 77.4 (0.6) | 69.8 (0.6) | 81.8 (0.9) | 73.69   |

---

> > ### Comment · Reviewer_U3kd · 2023-08-15
> > **Thank you for the rebuttal.**
> >
> > First of all, thank you for providing the discussion about the points I mentioned.
> >
> > ---
> > **[W1] Novelty (comparison to [1])**
> >
> > - Our paper's focus: I do not agree with this claim. Even though the authors did not intend the same effect of [1], the loss function is almost the same. I think this significantly limits the novelty of this work.
> >
> > - We do not consider views & We filter the graphs based on the distances: As I mentioned in the original review, I'm aware of these slight differences. However, at least for me, this makes no much difference. As far as I understand, [1] repels the representations of distant molecules in terms of ECFP, while this work repels the representations of distant molecules in terms of PI. If this is not true, please correct me.
> >
> > **[W2 &W5] & [W4] "Flexibility" of TDL**
> >
> > - "flexibly" is intended to convey the fact that it clearly adds a novel form of regularization which is likely effective: I do not agree with this claim. Similar to the contrastive learning objectives in conventional molecular representation learning (e.g., GraphCL, JOAO), TDL can also applied as a standalone objective (there is no specific reason for TDL to be only utilized as a regularization). If the authors want to claim that TDL is effective, the tables should have been designed as GraphCL vs JOAO vs TDL vs other molecular pretraining methods.
> >
> > I think my concerns have not been resolved, and I would like to keep my score. Please let me know if I have misunderstandings. Thank you.

---

> ### Author Response · Authors · 2023-08-15
> **Thank you for getting back to us!**
>
> We indeed believe that there is some misunderstanding and try to clarify.
>
> **W1 Novelty in Comparison to [1]**
>
> - We focus on **a rather different loss function**\
> Please see the denominator, we apply a "filter" rather than weights. Subtle changes can have huge impact in ML and entire papers have been written about this kind of seemingly small adaptations. In the appendix, ablation experiments show that *this is the appropriate method for topological distances* (vs. ECFP, as used in [1]), which is the area we want to study in the context of SSL.
>
> - **The nature of our work is different**\
> Note that [1] appeared in a chemistry journal and **the analysis in [1] focuses on aspects such as explainability which are most interesting for chemists, while our paper has a clear technical, ML focus** and investigates aspects which are relevant in this field:
>    - we introduce *persistent homology, a mathematical method with well-known theory*, to molecular SSL
>    - we consider *various, popular baselines to prove the generality* of our work
>    - we provide *extensive linear probing* experiments
>    - we show considerable improvements in the latter and in small data settings for all baselines which, to the best of our knowledge, have *neither been considered nor obtained similarly in any related work on molecular SSL*
>
>
> - Citing the **reviewer guidelines**
> > Originality: Are the tasks or methods new? Is the work a *novel combination of well-known techniques? (This can be valuable!)* Is it clear how this work differs from previous contributions? Is related work adequately cited
> >
> We acknowledge that the latter two points were missing in the submission, but adding the citation and three sentences describing it will neither change the nature of our work nor its original contributions, which were clearly recognized by other reviewers.
>
> ---------------------------------------------------------
> **W4 "flexibly"**
>
> To resolve the concern that we are misrepresenting our method, we can certainly remove that word.
>
> ---------------------------------------------------------
> **W2 & W5 Standalone TDL**
>
> As any other loss function, TDL can be applied as standalone objective. Yet, as the paper's title points out, our submission's instantiation (without views) and evaluation **focus on investigating how PH can *improve* existing methods based on its complementary nature**. The suggested study using views is  interesting follow-up work which definitely should consider the various other potential benefits of PH.
>
> ---------------------------------------------------------
> Lastly, we note that "Reject" means, "a paper with technical flaws, weak evaluation, inadequate reproducibility and incompletely addressed ethical considerations."
>
> We completely do not understand this rating and hope that the above delimitation helps clarifying our work's contribution.
>
> **Thank you very much for getting back to us with the remaining concerns and for being open to further discussion!**

---

> > ### Comment · Reviewer_U3kd · 2023-08-20
> >
> > Thank you for the detailed response.
> >
> > I have a further question. Even if I admit that this method is intended to **focus on investigating how PH can improve existing methods based on its complementary nature**, then I think that the effectiveness of this method should be verified by the compositions of other methods. Every molecular representation learning method has its own perspective to improve the performance. For example, GraphCL can be viewed as a method that **focuses on how discriminating similar and different view can improve existing methods based on its complementary nature**, upon ContextPred. Therefore, in my point of view, Table 1,2,3,4 do not provide a meaningful insight about **how PH can improve existing methods**. In other words, there is no baselines compared in those tables and the comparison should be "ContextPred + TDL" vs "ContextPred + GraphCL" vs compositions of other methods.
> >
> > Therefore, I think my concerns are not fully resolved and I would like to keep my score. Thank you.

---

> ### Author Response · Authors · 2023-08-21
> **Author Response**
>
> Thank you for providing more details, we'll try to clarify.
>
> We decided to focus on the distances between **samples**, since **this dimension of the problem is completely neglected by existing CL methods focusing on views**. We believe this aspect bears most novelty. The results in Tables 1,2,3,4 target this topic.
>
> We agree that it is likely promising to also apply PH to create views and we are actually focusing on this topic in our follow-up research. Reviewer XtrW mentioned this direction as well, as a "completely different approach" that deserves further research. There are many more potentially interesting topics which could exploit the complementary nature of PH, so we had to choose one to start with.
>
> It is very unlucky if our writing caused confusion.
> - If "complementary nature" is misleading in your opinion, we are definitely open to change the wording.
> - If GraphCL vs. GraphCL + TDL does not show that TDL "improves" GraphCL, we can change "improve" to an alternative notion (maybe "complements") which better describes our methods goals.
>
> We hope that the confusion which seemingly came from two words does not impact the recognition of our actual contributions.
>
> ------------------------------------------------------
>
> In order to address your concerns, we ran "ContextPred + GraphCL" to provide some more comparison, we tried "ContextPred + TDL" before. However, combining such very different kinds of models turned out challenging and the results would need further and careful tuning for both models to provide real insight. This is probably also why this kind of model combinations is usually not considered in the literature.

---

> > ### Comment · Reviewer_U3kd · 2023-08-21
> >
> > Thank you for detailed response to alleviate my concern.
> >
> > Indeed, in contrast to the authors' claim, several CL methods focusing on view studied about the distances between samples. For example, [1] utilizes "hard negative" samples to improve the performance of molecular representation learning. This paper and [1] introduce opposite objective: This paper discriminates "distant" molecules while [1] discriminates "nearby" molecules. If these two approaches were carefully analyzed in the manuscript, I might have agreed on the novelty that the authors argued. However, in the current manuscript, the claimed novelty seems not well-supported.
> >
> > [1] Molecular Contrastive Learning with Chemical Element Knowledge Graph, AAAI 2022

---

> ### Author Response · Authors · 2023-08-22
> **Author Response**
>
> This is a fair and very critical point. Our above statement is too absolute. We are sorry, this happened in the heat of the moment and was truly without intention.
>
> **Related Work.** In fact, [1] mentioned in your initial review already considers samples to some extent. In our updated related work section, next to the works from the reviews, we also had incorporated some other works using hard negatives and correlation (i.e., in SSL more generally). Note that the ToDD paper, to which we provide a detailed delimitation in our submission, uses hard negatives as well.
>
> **Our approach is different.** The ToDD paper and some others use hard negatives in the supervised setting and hence have label information available to, in a sense, safely select the samples. In the paper mentioned and in the other SSL works we found, hard samples are used to build the batch and hence to shape the nominator in the CL loss (i.e., the distances between views). In contrast, we have pairs of samples in the denominator and explicitly model the distance between those. We can definitely add more discussion to support our novelty. In our preliminary experiments, we weighted samples in the nominator of the regular, view-based CL loss, similar to [1]. But TDL showed better performance.
>
> Furthermore, we hypothesize that, in our setting, pushing away similar samples might be critical conceptually since they may still have similar properties. Therefore we focus on pushing away truly negative samples. The closest related work in this sense is probably [1], and we have provided a detailed written and experimental comparison to that in the rebuttal.
>
> **Our paper demonstrates novelty** in showing that TDL suits PH in that it can greatly complement a great number of CL approaches and improve performance in various interesting settings.
>
> Selecting hard negatives based on PH would certainly be another possible application of PH in SSL, similar to the views based on PH, discussed previously. We will add this to the more general discussion about the potential of PH for molecular SSL.
>
> [1] Improving Molecular Contrastive Learning via Faulty Negative Mitigation and Decomposed Fragment Contrast, Wang et al., JCIM 2022

---

### Official Review · Reviewer_9rP5 · 2023-07-08

**Soundness:** 2 fair
**Presentation:** 2 fair
**Contribution:** 2 fair
**Rating:** 5
**Confidence:** 4

**Summary:**

The paper proposes two approaches to leverage topological information (obtained from persistent homology) for molecular representation learning in a self-supervised setting. The first (TAE) uses an encoder-decoder architecture whose decoder aims to recover topological fingerprints. The second approach (TDL) consists of a contrastive loss based on the similarity between topological fingerprints. The latter is combined with existing contrastive learning methods. Experiments on linear probing and downstream prediction tasks show the efficacy of the proposals.


**Strengths:**

- Ablation studies: There is a substantial number of experiments and ablation studies.
- I like the simplicity of the proposed approach.
- Flexibility: TDL can be combined with most SSL approaches.

**Weaknesses:**

- Overall, I believe the paper provides limited insight to support the proposals. Also, it does not discuss which structural information the proposed approach captures but not existing methods. From a conceptual level, we know that 1-WL GNNs cannot capture information even from simple homology (e.g., number of independent cycles of a graph). Thus, TAE has inherent limits/failures. In other words, the topological information we loose after pushing a graph through a GNN (which would be captured by TDA) cannot be recovered from GNN embeddings.
- Results on downstream tasks: Based on Table 4, the gains from TDL look marginal. The gain is less than one standard deviation from the base model for many datasets.
- Incorporation of domain knowledge: The claim that the proposal allows for incorporating domain knowledge seems overstated. The basis for such a claim comes from the choice of the filtration function. However, it is unclear how different filtration functions affect the topological embeddings --- thus, domain experts cannot leverage their knowledge to choose the filtration functions.
- TAE vs. TDL...which one should we use? The paper says that "TAE, which we developed for comparison purposes only..." (line 283). I am unsure whether TAE should be introduced as a main contribution or as a baseline (in the experiments) for assessing the feasibility of learning the topological fingerprints with a simple architecture.


**Questions:**

1. Could the authors elaborate on why the fact that 'the Euclidean distance between PIs is stable with respect to the 1-Wasserstein distance between PDs' (line 111-112) is relevant here? Don't we want stability wrt to the input graphs?
2. What does the paper mean by calibrated distances?
3. Can the proposed methods be extended to employ learnable filtration functions?
4. A significant part of the experiments is devoted to showing the alignment between the learned molecular representations and the topological fingerprints. Isn't it naturally expected from the proposed design (e.g., additional loss term)?
5. Could the authors elaborate more on the fact that TAE can capture inter-molecule relationships if they learn the PIs? Can't GNN embeddings also learn important structural information and be stable?
6. Have the authors considered applying only TDL without other loss terms from CL methods? If yes, how well does it work?
7. The paragraph 'linear probing' (line 281) says 'we evaluated extensively using MLPs on the representations of the pre-trained graph encoders...'. Shouldn't this employ linear models instead of MLPs?
8. Duplicated text, e.g.,
    - Lines 1-5 --> 20-23
    - Lines 24-31 --> 199-206
    - Lines 5-6 --> 37-38
9. Typos:
    - GraphCL propose (line 32)
    - JOAO extend (line 33)
    - and to exploit (line 65)
    - TDL provides as a form of regularization (line 233)

**Limitations:**

The authors mention limitations in the main paper (section 5).

---

> ### Author Rebuttal · Authors · 2023-08-09
>
> **Thank you for the very detailed feedback!**
>
> We reformulated the particular benefits of PH for molecular SSL below and also adapted the paper. We hope that this clarifies the initial confusion about our contribution, please let us know in case further details are needed!
>
> W2 and W3 are addressed in the global reply. And for W1 and W3, please also see the reply to recC, Q5.
>
>
> **W1 & W4 Clarification on TAE**
>
> Observe that TAE is encouraged to directly learn the PI via the objective function, hence it does not necessarily have to reproduce the algorithmic procedure. To verify its ability, we ran preliminary experiments applying the pretrained, not finetuned TAE for PI prediction on the evaluation data. The Pearson correlation coefficients between the predicted and real PIs show that it approximates the latter PIs fairly well:
>
> |      | Tox21  | ToxCast | Sider  | ClinTox | MUV    | HIV    | BBBP   | Bace   |
> | ---- | ------ | ------- | ------ | ------- | ------ | ------ | ------ | ------ |
> | TAE  | 0.8572 | 0.7744  | 0.5939 | 0.8642  | 0.9044 | 0.7359 | 0.8660 | 0.8514 |
>
> Moreover, TAE is intended as simple and straightforward baseline for PIs in SSL, in the way the area uses the baselines of Hu et al. as context for interpreting results. In particular, since TDL does not directly encourage the model to learn PIs but rather the relations between them, the comparison to TAE allows us to evaluate the effectiveness of this rather abstract goal. Nevertheless, note that our comparison is only coarse since it is based on numbers (i.e., instead of on the actual predictions). A similar, even closer comparison could be drawn by considering regular CL using the PIs as embeddings, as outlined by Reviewer XtrW.
>
>
> **Q1 & Q5 Relevance of Stability of Topological Fingerprints, and Relation to TAE**
>
> Stability w.r.t. the input graphs is definitely the goal, but we need an appropriate metric for the data space. There is no unique such metric for graphs, and we are only aware of few such works (e.g., a recent paper proposes a custom tree-mover's distance [1]). Our paper's hypotheses are that
>
> - the stability certain topological fingerprints offer is a mathematically grounded, well-studied, and efficient proxy for the stability w.r.t. graphs, which we can use complementary to existing approaches;
> - stable representations support the learning of a well-structured embedding space, which is particularly important for SSL;
> - the fact that they are generic in the filtration function gives domain experts the opportunity to flexibly inject their knowledge, which particularly suits molecular representation learning (see also the references mentioned by Reviewer recC).
>
> Altogether, this motivates us to study persistent homology in the context of SSL over molecular graphs.
>
> Assuming TAE learns PI-based representations sufficiently well, then the distances ("relations") between its embeddings reflect the ones between the corresponding input graphs in terms of the 1-Wasserstein distance between their persistence diagrams.
>
> As for example [1] show, stability can indeed be defined for GNN embeddings. Our paper is intended to complement this research by investigating how we can leverage the body of existing works in persistent homology in the context of graph representation learning and, in particular, in molecular SSL.
>
>
> **Q2 What does the paper mean by calibrated distances?**
>
> TDL intends to "adjust" the distances between the embeddings such that they better reflect the ones between the PIs. In this sense, they get calibrated. We are aware that the notion is not perfect but did not find a better one. We are definitely open for suggestions! For now, we added this explanation to the paper.
>
> **Q3 Can Filtration Functions be Learnt?**
>
> While this should be possible in general, it might be expensive in an SSL setting since the topological fingerprints then have to be reconstructed in each epoch. Furthermore, in the context of our TAE and TDL, the objective functions are not clear.
>
> **Q4 Why Experiments showing Architecture Alignment?**
>
> While we indeed designed our architecture as carefully as possible, we do not think that such design intentions or even theoretical architecture guarantees have to necessarily translate into practice (e.g., GIN has been shown to be very expressive, but there are datasets where the conventional GCN is superior [2]). Therefore, we consider these experiments showing alignment of the architecture as considerable contribution, in particular, also because the relation to the topological fingerprints offers various topics for future investigation (see also the response to Reviewer bcsi).
>
> **Q6 What about TDL w/o other Losses?**
>
> Given that TDL does not incorporate the regular, proven views used in CL, we do not recommend this setting. *We did some such experiments, interestingly obtaining quite good performance, now* (see U3kd, Q1 & Q3 TAE + TDL?). Note that regular CL views could certainly be constructed based on PIs or other topological fingerprints. In our initial study, we chose a different, more novel focus, but we are investigating alternative options now.
>
>
> **Q7-Q9 Minor Comments**
>
> Thank you for checking on this level of detail! The "MLP" in the context of linear probing is indeed a mistake, we used a simple linear layer. We also fixed the remaining items.
>
> ----------------------------------------------------------------------------------------
> [1] Chuang, et al. "Tree Mover's Distance: Bridging Graph Metrics and Stability of Graph Neural Networks." Advances in Neural Information Processing Systems 35 (2022): 2944-2957.
>
> [2] Dwivedi et al. "Benchmarking graph neural networks." JMLR (2023).

---

> > ### Comment · Reviewer_9rP5 · 2023-08-18
> >
> > Thank you for taking the time to answer my questions and comments. While some of my concerns were cleared, I still believe the paper provides little insight to support the proposal. In an effort to address my conceptual question, the authors provide a table showing correlations between PIs and the GNN approximations (varying from 0.59-0.90). This is insufficient and does not strengthen the motivation for the proposal. Overall, I think the contribution is not theoretically grounded nor builds upon solid claims.
> >
> > Also, I believe the claim that the proposal allows for incorporating domain knowledge seems overstated. In their reply, the authors have run additional experiments with a "stronger filtration function" with "more domain knowledge". What does stronger filtration mean here? --- Is it capable of capturing topological information that the previous one couldn't? What does a filtration function say to domain experts, and how can they choose the best filtration?
> >
> > I acknowledge that I have read the other reviews and authors' responses. Since some of my concerns were alleviated, I am increasing my score from 4 to 5.

---

> ### Author Response · Authors · 2023-08-15
> **Question by Authors**
>
> Dear Reviewer 9rP5,
>
> Since your initial review pointed out several detailed questions and concluded with a slightly negative overall rating, please let us know in case there are remaining concerns which we can address.
>
> Thank you again for providing that much feedback. Your recognition of the paper's strengths is very valuable!

---

> ### Author Response · Authors · 2023-08-18
> **Thank you!**
>
> Thank you for acknowledging our rebuttal and for getting back to us!
>
> We try to clarify below.
>
> **C1 Theoretical Grounding of Contributions and Claims**
>
> As described in the paper, TAE is intended as simple, straightforward baseline, without specific theoretical grounding. Topological fingerprints have shown promising results in several past works, and TAE is just trained to predict those. In this way it also likely loses aspects of the molecules, which are not captured in the topological fingerprints.
>
> TDL is theoretically grounded in that our objective is based on the stability of the topological fingerprints. Molecules which have more similar PDs are moved closer to each other in the embedding space. Since the training reduces the loss function, as stated in your review, our design models the theoretical contribution to some extent. Our empirical evaluation tries to complement that. Since the representations after pre-training capture the learnt knowledge clearest we have paid special focus on linear probing and small datasets, and we see strong improvements there. We think the distance probing experiments are strong since they show that the embeddings make the model capture distance to a certain extent, which is an important capability (see also the rebuttal for bcsi).
>
> However, we also applied k-NN on the PIs (i.e., on the raw fingerprints, not on embeddings; not presented in the paper/rebuttal so far), to **verify that the relations between the PIs in fact capture some information and can be used for supervision and hence to build upon solid claims**. Maybe this is a more direct demonstration, in the way you had in mind. We see that they seem to capture nearly as much - and likely different - knowledge as ECFP.
>
> |      | Tox21 | ToxCast | Sider | ClinTox | MUV  | HIV  | BBBP | Bace |
> | ---- | ----- | ------- | ----- | ------- | ---- | ---- | ---- | ---- |
> | PI   | 58.8  | 51.1    | 58.7  | 50.9    | 50.2 | 64.8 | 55.2 | 75.5 |
> | ECFP | 63.8  | 54.6    | 59.1  | 50.7    | 54.0 | 68.1 | 59.3 | 77.0 |
>
>
> **C2 Meaning of "Stronger" Filtration**
>
> We in fact missed to give more details about that.
> Our initial filtration function only used atom symbols to construct regular PIs (to allow for  a pure technical comparison with others), while this one
> 1. Consider **various types of information**
> - a weight filtration to express bond strength in the compounds. Single bond has weight 1, double bond has weight 2, triple bond has weight 3, and finally aromatic bond has weight 4 on the edges.
> - a sublevel filtration on partial atomic charges and
> - a sublevel filtration on atomic mass
> 2. For each of those three, we do not simply consider the given filtration, but **construct a 2D filtration by filtering**: in one dimension according to the above information and in the second dimension according to a VR filtration capturing the distances between atoms.  (Fig. 3 in [1] illustrates this kind of multi-dimensional filtration)
> 3. Lastly, the **3 2D PIs are concatenated**, and we compute distances based on the combination of PIs
>
> **C3 What does a filtration function say to domain experts, and how can they choose the best filtration?**
>
> Filtration functions are techniques used in models, similar to how we use kernels in ML. The choice is based on the data and the application scenario. In SSL, we hypothesize, that more generally applicable filtrations are likely more effective. But this will have to be validated in practice, of course. In fact, the filtration from [1] captures such basic knowledge and showed good performance in both their work and our experiments, hence it provides a good starting point. Moreover, we believe the inclusion of 3D information might turn out to be helpful as well.
>
> **Example.** The BBBP dataset focuses on the assessment of compounds' blood-brain barrier (BBB) penetration, it is observed that polar-related descriptors tend to exhibit inverse correlations with BBB permeability [2]. And Partial atomic charges are a measure of the degree of electronegativity of an atom in a molecule and can indicate the polarity of atomic interactions. Therefore, partial atomic charges could potentially serve as filtering functions in this dataset. The ToDD filtration includes them, and our results show that the partial charges here may indeed change the picture. Since there is other information involved, we cannot draw direct conclusions; yet the increase on BBBP is much higher than on all other datasets (see .pdf).
>
> |                       | BBBP       |
> | --------------------- | ---------- |
> | TAE                   | 67.5 (1.1) |
> | TAE (ToDD filtration) | 70.4 (0.8) |
>
> [1] ToDD: Topological Compound Fingerprinting in Computer-Aided Drug Discovery, NeurIPS 2022.
>
> [2] Jiang, Dejun, et al. "Could graph neural networks learn better molecular representation for drug discovery? A comparison study of descriptor-based and graph-based models." *Journal of cheminformatics* 13.1 (2021): 1-23.
>
> (edited the example 08/21)

---

### Official Review · Reviewer_bcsi · 2023-07-11

**Soundness:** 3 good
**Presentation:** 3 good
**Contribution:** 3 good
**Rating:** 7
**Confidence:** 2

**Summary:**

This paper explores self supervised learning in the context of molecular representation, specifically based on persistent homology. The paper proposes an autoencoder to demonstrate the general representational power of PH and a contrastive-learning-based loss that can be applied to existing SSL approaches. The proposed approach is evaluated for molecular property predictions, showing improved representations and predictive power compared to baselines across different tasks. The claim is that the new loss function enhances baseline performance particularly with small datasets.

**Strengths:**

The paper is well written and the idea is novel and interesting.

**Weaknesses:**

- Given the technical nature of PH, and its origin in the domain of topological data analysis, a more mathematical foundation of the methods in the paper would be desired.

**Questions:**

- Since PH naturally offers multiple data views, I wonder whether the learned representations can be more explainable, that is, not just focusing on atom based attribution but also having explanations of more global features/subgroups, for instance, along the lines of
Bertolini et al. "Beyond Atoms and Bonds: Contextual Explainability via Molecular Graphical Depictions".

**Limitations:**

The authors describe the limitations of their approach.

---

> ### Author Rebuttal · Authors · 2023-08-09
>
> **Thank you for the interesting comments!**
>
> The paper does not explicitly discuss the points mentioned in the review because these are challenging topics in themselves, but they nicely underline the future research potential and we are happy to discuss them further.
>
> **W1 More mathematical foundation would be desired.**
>
> We agree on that and definitely plan to investigate the learnt embeddings from a theoretical viewpoint, but we consider that out of the scope of this current, initial work on PH in the context of molecular SSL.
>
> It is not straightforward how to leverage the topological distances to obtain stability and generalization results, this deserves deeper study.
> A second topic which we address empirically is the increase in the rank of the embeddings. However, there we observe a strong dependence on the baselines, which will make the study more challenging. Moreover, this direction is only lately studied in SSL in general, with measurement methods presented as recently as at ICML 2023 [1].
>
> We sincerely thank the reviewer for recognizing that this kind of studies is beyond the focus of our paper and for not letting it influence the score too negatively.
>
>
> **Q1 Explainability**
>
> We see at least two ways in which topological features may offer particular advantages. First, the filtrations can be designed based on arbitrary domain knowledge, and this knowledge may indeed be re-discovered in the embeddings if both the filtration and the embedding are chosen carefully. For instance, there are certain kinds of porous crystals called MOFs whose pore diameters can be captured by using a combination of VR filtration and PIs [2]; in a nutshell, the filtration iteratively "aggregates" the pore-delimiting atoms, a novel topological structure appears once the process is finished, and the length of the aggregation is recorded in the persistence diagram (this example uses 2D topological voids and hence goes beyond the graph homology we introduce, but we think it is very illustrative). Note that for this kind of analysis the model has to be encouraged to learn the underlying topological fingerprint, e.g., in the way TAE learns PIs.
>
> Our CL-based approach uses the fingerprints in a different way, to retain the overall structure of the PH-based embedding space, in terms of distances. This offers a different angle of explainability, reflecting the relations/similarity between molecules. We believe that this is a unique feature of PH-based fingerprints and particularly interesting for SSL in that it directly supports the main goal of pre-training, learning a comprehensive embedding space. Also model calibration (important for interpreting the results) is critically relying on distance awareness [3].
>
> -----------------------------------------
>
> [1] Garrido, et al. "Rankme: Assessing the downstream performance of pretrained self-supervised representations by their rank." ICML, 2023.
>
> [2] Krishnapriyan, et al. "Machine learning with persistent homology and chemical word embeddings improves prediction accuracy and interpretability in metal‑organic frameworks" Scientific reports 11.1 (2021): 8888.
>
> [3] Liu, et al. "Simple and principled uncertainty estimation with deterministic deep learning via distance awareness." NeurIPS, 2020.

---

> > ### Comment · Area_Chair_R4nR · 2023-08-18
> >
> > Dear reviewer,
> >
> > Thanks for supporting the review process. Please briefly acknowledge the rebuttal by the authors and ask for additional clarifications if required.
> >
> > Best,\
> > Your AC

---

### Author Rebuttal · Authors · 2023-08-09


**We thank all reviewers for the very fair, detailed, and constructive feedback!**

We address all comments below and are happy to provide additional information if needed.

---------------------------
**G1 Summary of Additional Experiments Suggested by Reviewers**

- **Technically similar approach** ([1] suggested by U3kd). Performance well below GraphCL+TDL when it is pre-trained over the same data (~2M molecules rather than the ~10M considered in [1]), and TDL yields improvement.
- **Standalone TDL** (9rP5 Q6). Though not recommended, achieves a competitive average ROC-AUC of 72.02, *without using views*.
- **Recent AD-GCL** (U3kd Q2, XtrW W2). TDL increases average performance from 72.67 to 73.21. Our baselines+TDL often outperform AD-GCL in our additional experiments by large margins (e.g., linear probing).
- **w/ Stronger Filtration, ToDD [6]** (9rP5 W1, XtrW W3, recC Q5). Convincing improvements, *across several CL baselines*:
    * Linear probing: average performance raised to >= 67 (vs. <= 64 w/o TDL), which is comparable to fine-tuned GIN
    * Fine-tuning: improvement between 1.3 and 2.0 for all models
    * Low-data fine-tuning: performance improvements of up to 10%

**Please see the attached pdf.** We report some results and additional side experiments in the reviews.

---------------------------
**G2 Comparison to SOTA** (U3kd Q2, XtrW W2)

The SEGA paper appeared on arXiv on May 8 and the code is not yet available. We tried to run Mole-BERT before for quite some time but we did not succeed, and all issues on its repository have gotten deleted so far, without having been addressed.
We now ran TDL on top of AD-GCL and see good improvement (see attachment). Moreover, the inclusion of AD-GCL into the other experiments nicely *highlights the complementary nature and potential of TDL: overall, it makes the baselines outperform AD-GCL, often by large margins*.

---------------------------
**G3 Power and Complementary Nature of PH-based Embeddings, compared to regular GNN embeddings or ECFP** (9rP5 W1&W3, U3kd W3, XtrW W3, recC Q1&Q5)

- **References.** In the submission we only mentioned the work about molecule representation learning using persistent homology which is closest to ours and omitted the majority of works in the area since the technical focus of our paper is on SSL, a specific setting with its own challenges. However, as suggested by Reviewer recC and as it is shown in the reviews, PH is less known in the ML community and *we can back our claim that it represents a proven and powerful method for representation learning over molecules according to domain scientists*. We will add more discussion about this to the appendix and hope that we thereby address 9rP5 W3 (partially), U3kd W3, and recC Q1.
- **Power of Filtration Function.** In the submission, we focused on a most simple filtration function based on atom symbols, which does not use more domain knowledge than related works. This allows us showing that TDL adds a novel dimension of knowledge (i.e., the explicit distance regularization) even in this scenario. We also introduced the most simple filtration in the theory part, to convey the overall idea. *We did not intend to recommend a particular filtration, TDL is generic there*. To provide more support for our proposal of using PH (9rP5 W1, XtrW W3, recC Q5), *we report hopefully convincing results in the attachment using a stronger filtration* function including more domain knowledge, the one proposed in [6].

---------------------------
**G4 Relevance of Fine-Tuning Experiments** (9rP5 W1&W2, XtrW W1)

Please note that "the experimental results" should not be equated with Table 4 alone. We do not consider the fine-tuning experiments to show our method's effectiveness best. TDL shines in other scenarios.

Furthermore:
- In drug discovery, downstream benchmarking results rarely translate into practice which is getting recognized more and more recently (e.g., see the ICML 2023 panel on the topic [1] or [2]). Hence, the *generally* minor model differences over Moleculenet should be interpreted with care.
- SOTA works in SSL in other domains sometimes do not consider fine-tuning experiments at all [3, 4], since the data contains various forms of additional confounding factors (e.g., label distribution, dataset balance) which make it hard to get an insight into the actual influence of the SSL embeddings. Generally, fine-tuning experiments are considered as one possible label-based evaluation protocol, often only considered after linear probing etc. [5].

The Moleculenet benchmark used with scaffold split represents a challenging setting, which gives good estimates about model performance in certain downstream scenarios. However, we think that graph SSL should extend the experiment setting introduced by Hu et al. in 2020, which solely relies on fine-tuning, and evaluate more broadly.

TDL *significantly* improves a range of well-known and more recent graph SSL approaches on Moleculenet
- in linear probing and
- in low-data experiments on subsets of the benchmark.

Our additional results using a stronger filtration method even improve upon the numbers reported in the submission and also show that PH may lead to considerable increases in fine-tuning.


---------------------------
[1] ICML 2023 Panel: Fostering the Development of Impactful AI Models in Drug Discovery.

[2] Tossou et al. "Real-World Molecular Out-Of-Distribution: Specification and Investigation." ChemRxiv.

[3] Devillers, et al. "EquiMod: An Equivariance Module to Improve Visual Instance Discrimination." ICLR 2023.

[4] Ermolov, Aleksandr, et al. "Whitening for self-supervised representation learning." ICML 2021.

[5] Balestriero, Randall, et al. "A cookbook of self-supervised learning." arXiv preprint arXiv:2304.12210 (2023).

[6] ToDD: Topological Compound Fingerprinting in Computer-Aided Drug Discovery, NeurIPS 2022.

---

### Decision · Program_Chairs · 2023-09-21

**Decision:**

Accept (poster)

**Comment:**

This paper presents a novel way for self-supervised molecular representation learning using novel algorithms from computational topology. All reviewers agreed about the significance of the topic and the relevance of the proposed method. While one reviewer raised some concerns about the novelty (`U3kd`), I find no such faults to be part of the current write-up. In addition, the authors provided a strong and comprehensive rebuttal that served to clarify numerous relevant aspects. I am thus happy to endorse this paper for publication. The authors are **strongly encouraged** to make use of the suggestions by reviewers and integrate the additional experiments mentioned in the rebuttal. Moreover, a revision in terms of improving the clarity would be beneficial in order to make this work available to a broader audience. I trust the authors to address these points in their revision.